# Moderate white light exposure enhanced spatial memory retrieval by activating a central amygdala-involved circuit in mice

MengJuan Shang[1,7], MeiLun Shen[1,7], RuoTong Xu[2], JingYu Du[2], JiMeng Zhang[3], Ding OuYang[2], JunZe Du[2], JunFeng Hu [1], ZhiChuan Sun[4], BingXia Wang[1], Qian Han[1], Yang Hu[2], YiHong Liu[2], Yun Guan[5], Jing Li[1], GuoZhen Guo[1,6 ✉] & JunLing Xing [1,6 ✉]

Light exposure can profoundly affect neurological functions and behaviors. Here, we show that short-term exposure to moderate (400 lux) white light during Y-maze test promoted spatial memory retrieval and induced only mild anxiety in mice. This beneficial effect involves the activation of a circuit including neurons in the central amygdala (CeA), locus coeruleus (LC), and dentate gyrus (DG). Specifically, moderate light activated corticotropin-releasing hormone (CRH) positive (+) CeA neurons and induced the release of corticotropin-releasing factor (CRF) from their axon terminals ending in the LC. CRF then activated tyrosine hydroxylase-expressing LC neurons, which send projections to DG and release nor-epinephrine (NE). NE activated β-adrenergic receptors on CaMKIIα-expressing DG neurons, ultimately promoting spatial memory retrieval. Our study thus demonstrated a specific light scheme that can promote spatial memory without excessive stress, and unraveled the underlying CeA-LC-DG circuit and associated neurochemical mechanisms.

[1] Department of Radiation Biology, Faculty of Preventive Medicine, Fourth Military Medical University, Xi'an, ShaanXi 710032, China. [2] The Third Regiment, School of Basic Medicine, Fourth Military Medical University, Xi'an, ShaanXi 710032, China. [3] The Second Regiment, School of Basic Medicine, Fourth Military Medical University, Xi'an, ShaanXi 710032, China. [4] Department of Neurosurgery, Daxing Hospital, Xi'an, ShaanXi 710032, China. [5] Department of Anesthesiology and Critical Care Medicine, the Johns Hopkins University, School of Medicine, Baltimore, MD, USA. [6] Ministry of Education Key Lab of Hazard Assessment and Control in Special Operational Environment, Fourth Military Medical University, Xi'an, ShaanXi 710032, China. [7] These authors contributed equally: MengJuan Shang, MeiLun Shen. ✉email: guozhen@fmmu.edu.cn; xingjunl@fmmu.edu.cn

Light is a significant environmental factor that can affect important physiological functions, including mood, cognition, memory, and sleep[1–4]. Extensive research has demonstrated the acute effects of light exposure on learning and memory, which can be either facilitatory or inhibitory. For instances, bright light impaired BALB/c mice performance in the water maze, a common spatial learning test[5] and spontaneous object recognition task[6]. In contrast, applying the light before the learning task improved contextual fear memory consolidation[7], and enhanced behavioral responses to learned fear[8]. Notably, previous studies used different behavioral paradigms and light interventions, and the tests were conducted in various stages (encoding/training, reconsolidation, retrieving) of learning and memory. Nevertheless, how light exerts different modulatory effects on memory retrieval remains unclear. Moreover, findings from human studies have been insufficient for assessing the usefulness of light therapy for ameliorating dementia and memory disorders. Therefore, more mechanistic and translational studies are necessary[9].

Moderate level of anxiety is actually the result of human evolution. It develops under certain conditions of stress and can help us increase alertness and avoid danger[10,11]. Several hypothalamic nuclei such as the suprachiasmatic nuclei (SCN), the ventro-lateral preoptic nucleus (VLPO) and locus coeruleus (LC), as well as other brain nuclei, including the dorsal raphe (DR) and central amygdala (CeA), are involved in the effects of light on alertness[3,4,12]. These brain nuclei receive inputs from retina which detect light stimulation, and activation of these nuclei by bright light may cause anxiety[13,14]. In turn, anxiety can significantly affect a variety of brain functions, including learning and memory[15,16]. Accordingly, aversive stimulus (e.g., foot shock)[17] that can induce anxiety by itself should not be used in light-related memory behavioral paradigms, as it may confound the findings. In this regard, the Y-maze test has proved to be a suitable method to objectively evaluate the effect of light on short-term spatial memory, as it does not induce any stress[18].

Because different stages of memory (e.g., encoding, retrieval) are associated with distinct neural networks[19–22], the timing of light exposure may profoundly affect these processes. Here, we used complementary approaches to test the hypothesis that acute light exposure at proper intensity and timing may improve spatial memory retrieval without causing excessive anxiety in mice. Specifically, we examined the effects of different intensities of white light exposure during the memory training period in the Y-maze test on animals' spatial memory retrieval and anxiety level. The results were then compared to those induced by applying the light during the testing period. Importantly, we identified a neuronal circuit that mediates the behavioral changes induced by light exposure and further explored the underlying neurochemical mechanisms, using c-Fos staining, in vivo optical recording, virus tracing, chemogenetics, and transmitter sensor techniques.

## Results

**400 lux testing-period light exposure promoted spatial memory retrieval.** In the Y-maze test, mice were first exposed to different intensities of light (50–1200 lux) during the training period and then to 50 lux light during the testing period (Fig. 1a i), their time spent in the novel arm during the testing period was comparable among different mouse groups (Fig. 1a ii). However, when the light treatment was switched between the training and testing periods, the 400 lux group showed a significant increase in time spent in the novel arm, as compared to the 50 lux control group (Fig. 1b). Further analysis showed that there was no significant difference in the total distance traveled in the Y-maze test among different light intensity groups, indicating that change of light intensity did not affect locomotion (Fig. 1c). These findings suggest that 400 lux light treatment during the testing period, but not the training period, promoted short-term spatial memory retrieval. In contrast, the stronger 800 lux and 1200 lux groups showed significant decreases in time spent in the novel arm (Fig. 1b ii).

Stress can affect learning and memory[23,24]. Since bright light may cause stress and anxiety and thus affect memory, we further examined how different intensities of white light affect the anxiety level in mice. Compared to the 50 lux control, 400, 800 and 1200 lux light treatments significantly decreased animals' exploration time in the center zone in the open field test (Fig. 1d), and also reduced their time spent in the open arm in the elevated plus-maze test (Fig. 1e). These findings suggest that increasing the light brightness elevated the anxiety level in an intensity-dependent fashion. We postulate that the enhanced spatial memory retrieval induced by 400 lux light exposure during the testing period may have partially resulted from an "optimal" or "beneficial" level of stress which may reinforce memory retrieval. This notion is supported by our findings that 400 lux light also enhanced spatial memory retrieval in the novel object location test, another important paradigm for the assessment of short-term spatial memory (Supplementary Fig. 1). In line with our suspicion, the high-intensity light exposure (800, 1200 lux) induced excessive stress and anxiety, which would impair animals' exploration behavior and memory.

**400 lux testing-period light exposure induced neuronal activation in the CeA, LC, and dentate gyrus (DG).** Using c-Fos mapping, we examined brain neurons activated by 400 lux light exposure during the testing period of the Y-maze test. Particular attention was paid to brain regions known to be important to the regulation of alertness and cognition by light, as suggested by human functional magnetic resonance (fMRI) studies[25,26]. In both 400 lux and 50 lux groups, c-Fos positive (+) neurons were detected in multiple brain regions including the paraventricular nucleus of the hypothalamus, CeA, lateral hypothalamus, dorsal medial hypothalamus, posterolateral cortical amygdala nucleus, DG, precommissural nucleus, and LC (Fig. 2a i–h ii). Importantly, c-Fos expressions in the CeA, DG, and LC were significantly increased in the 400 lux group (Fig. 2b, f, h), as compared to the 50 lux control group (Fig. 2i). These findings suggest that 400 lux light-activated neurons in the CeA, LC, and DG, which may be involved in the process of enhanced spatial memory retrieval.

To further ascertain the activation of brain neurons by light, we conducted in vivo optical calcium imaging, allowing real-time neuronal activity monitoring[27]. The rAAV-GCaMP virus was injected into the CeA, LC, or DG in different groups of mice, and then an optical fiber was embedded in the injected region (Supplementary Fig. 2a). At 21 days after the virus injection, the expression of GCaMP6m was detected in the injected area (Supplementary Fig. 2c, d). In the Y-maze test, 400 lux light exposure during the testing period activated CeA neurons, as indicated by the increased fluorescence intensity (Supplementary Fig. 2e, n = 5). The heat map shows the light-induced increase in calcium signal in the CeA (Supplementary Fig. 2b). Similarly, increased calcium signal also occurred in the LC (Supplementary Fig. 2f–h) and the DG during 400 lux light exposure (Supplementary Fig. 2i–k).

**Identification of CeA-LC-DG circuit.** Previous studies showed that axons of CeA neurons project to the LC[28,29], and LC neurons may send projections to the DG[30]. To determine the presence of a

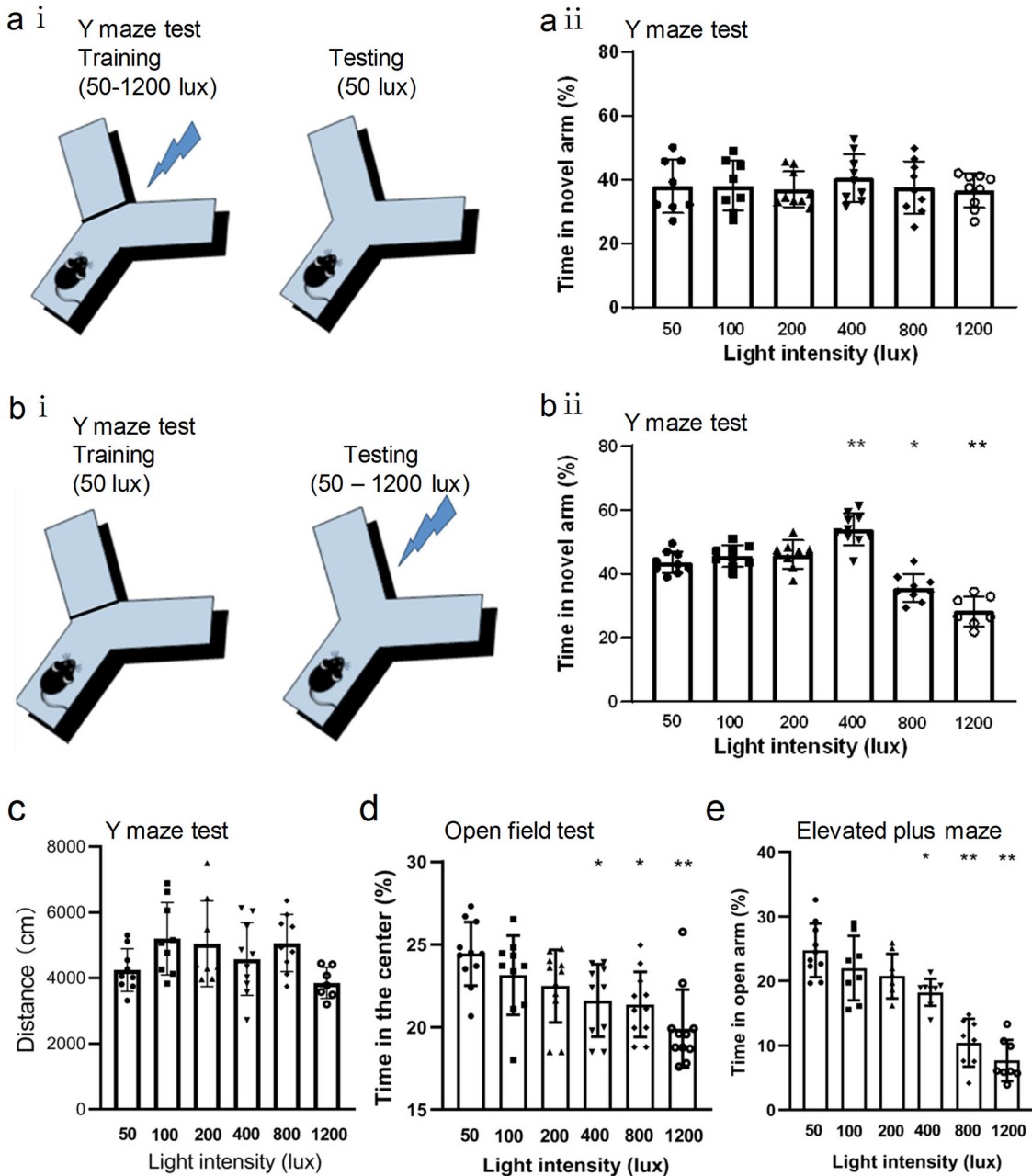

**Fig. 1 Effects of light exposure on spatial memory task performance and anxiety level in animal behavioral tests. a** (i) The schematic diagram of the Y-Maze test. Different intensities (50, 100, 200, 400, 800, 1200 lux) of white light were applied in the training period, and 50 lux light was applied during the testing period. **a** (ii) Percentage of the time that mice spent in the novel arm during the testing period in each group (non-parametric Kruskal-Wallis H Test, $n = 8$–9/group). **b** (i) In a separate study, different groups of mice received 50 lux light exposure in the training period, but were exposed to different intensities of light (50–1200 lux) during the testing period. **b** (ii) Percentage of the time that mice spent in the novel arm during the testing period ($n = 7$–10/group). **c** Distance of mice traveled in the Y maze test under different light intensity during the testing period (non-parametric Kruskal-Wallis H Test). **d** Percentage of the time that mice spent in the central area of the open field under the different intensities of light exposure (non-parametric Kruskal-Wallis H Test, $n = 7$–10/group). **e** Percentage of the time that mice stayed in the open arm of the elevated plus-maze under different light exposures ($n = 8$–10/group). Data are mean ± SD. *$P < 0.05$, **$P < 0.01$ vs. 50 lux group, one-way ANOVA with Dunnett-t multiple comparison test unless otherwise noted.

CeA-LC-DG circuit, we injected viruses for retrograde tracing into the DG and LC (Fig. 3a, b, d, e), and identified retrograde labeling in the LC and CeA, respectively (Fig. 3c, f). These findings confirmed direct projections from the CeA to LC, and from the LC to DG. We next used a transsynaptic tracing method which is based on a modified rabies virus to further confirm the presence of a CeA-LC-DG trisynaptic circuit. Neurons in the LC were infected by AAV expressing the rabies glycoprotein, and also by

EGFP tagger (a helper virus, Fig. 3g, h, i, and l) which helps rabies virus replication. Three weeks later, RV-EnvA-ΔG-dsRed (RV-DsRed) virus was injected into the DG (Fig. 3g, h, and j). These viruses can transfect the Helper+ LC neurons through retrograde transportation (Fig. 3m). LC neurons infected by both viruses will produce infectious RV-DsRed (Fig. 3n), which can propagate trans-synaptically to infect the upstream CeA neurons that send projections to the LC (ie., CeA-LC projection neurons, Fig. 3k). Using

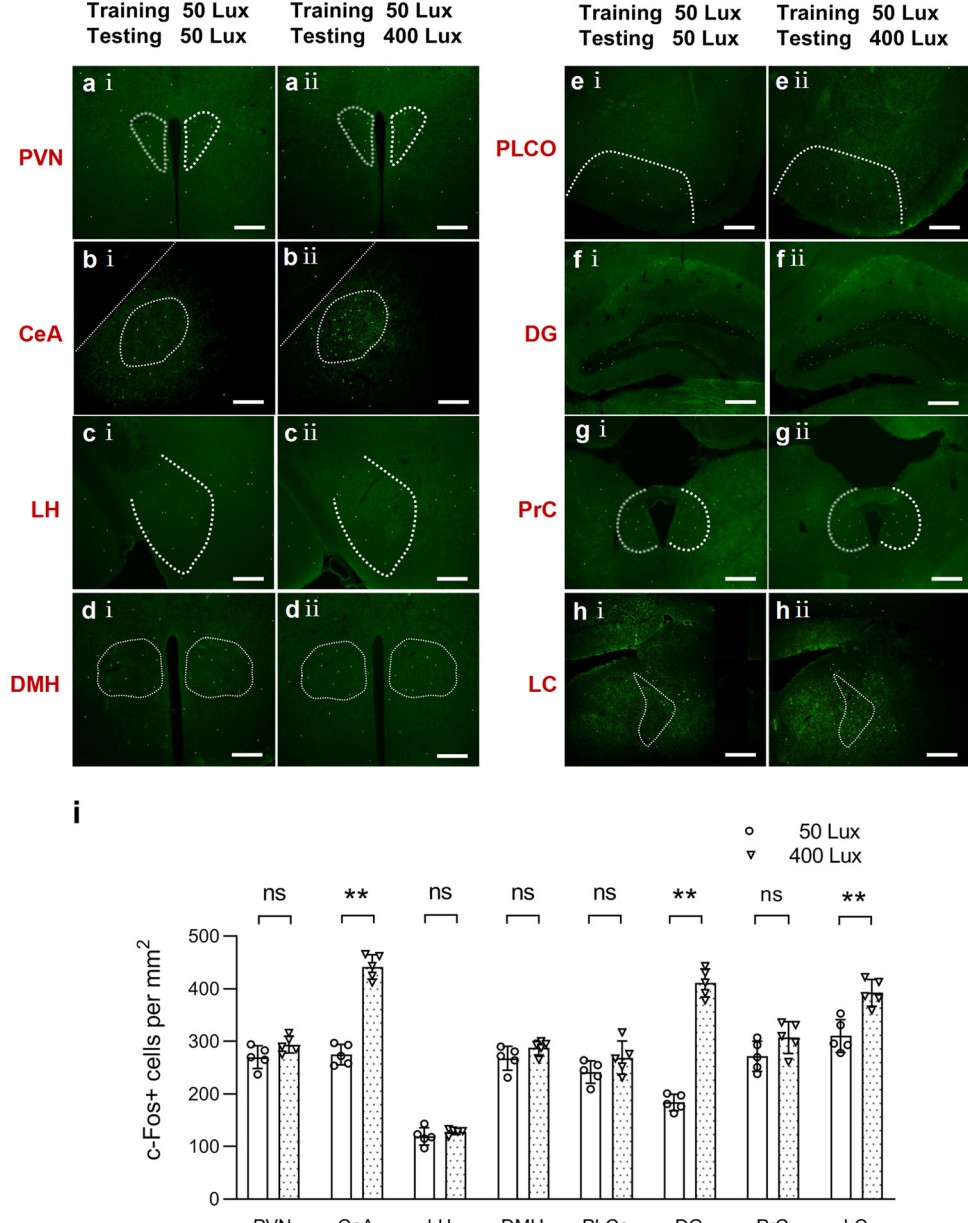

**Fig. 2 Expression of c-Fos in the brain after light exposure in the Y-maze test.** Representative images of immunofluorescence staining of c-Fos in different brain regions of mice after 50 lux light exposure during both training and testing period (**a** (i), **b** (i), **c** (i), **d** (i), **e** (i), **f** (i), **g** (i), and **h** (i)), and that after receiving 50 lux light exposure during the training period and 400 lux light exposure during the testing period (**a** (ii), **b** (ii), **c** (ii), **d** (ii), **e** (ii), **f** (ii), **g** (ii), and **h** (ii)). Scale bars: 200 μm. **i** Quantification of c-Fos+ neurons in each brain region. ns: not significant; **$P < 0.01$, $P > 0.05$ not significant vs. 50 lux group, two independent sample t-test, $n = 5$/group. CeA: central amygdaloid nucleus; LC: locus coeruleus; DG: dentate gyrus; PrC: precommissural nucleus; PLCo: posterolateral cortical amygdaloid nucleus; LH: lateral hypothalamus; PVN: paraventricular thalamus nucleus; DMH: dorsomedial hypothalamus.

this approach, we demonstrated for the first time direct projection from the CeA to LC, and then from the LC to DG.

To examine how light induces the activation of CeA neurons, we injected viruses into the CeA and eyeball for retrograde labeling. Our findings demonstrated projections from the retina to the medial amygdala (MeA), and then from MeA to CeA, which may present an anatomical pathway for light stimulus to activate the CeA neurons (Supplementary Fig. 3).

**The role of CeA-LC-DG circuit in light-induced facilitation of spatial memory retrieval.** Thus far, our findings have shown that 400 lux light exposure activated the CeA-LC-DG circuit. We next

took a chemogenetic approach to determine whether the promoted spatial memory retrieval depends on the functional activation of the CeA-LC-DG circuit by 400 lux light exposure. We expressed inhibitory chemogenetic receptors selectively in the CeA neurons that project to LC. To achieve this goal, the rAAV/hSyn-DIO-hM4Di-mCherry (rAAV-DIO-Gi-mCherry) virus was injected into the bilateral CeA (Fig. 4a–d), and rAAV-hSyn-Cre virus which can travel retrogradely to express CRE enzyme in the upstream brain region was injected into the bilateral LC (Fig. 4a, b). At 21 days after virus injection, CNO or saline was injected intraperitoneally (i.p.) at 30 min before the Y-maze test. Theoretically, CNO will selectively inhibit the activation of CeA neurons that project to the LC.

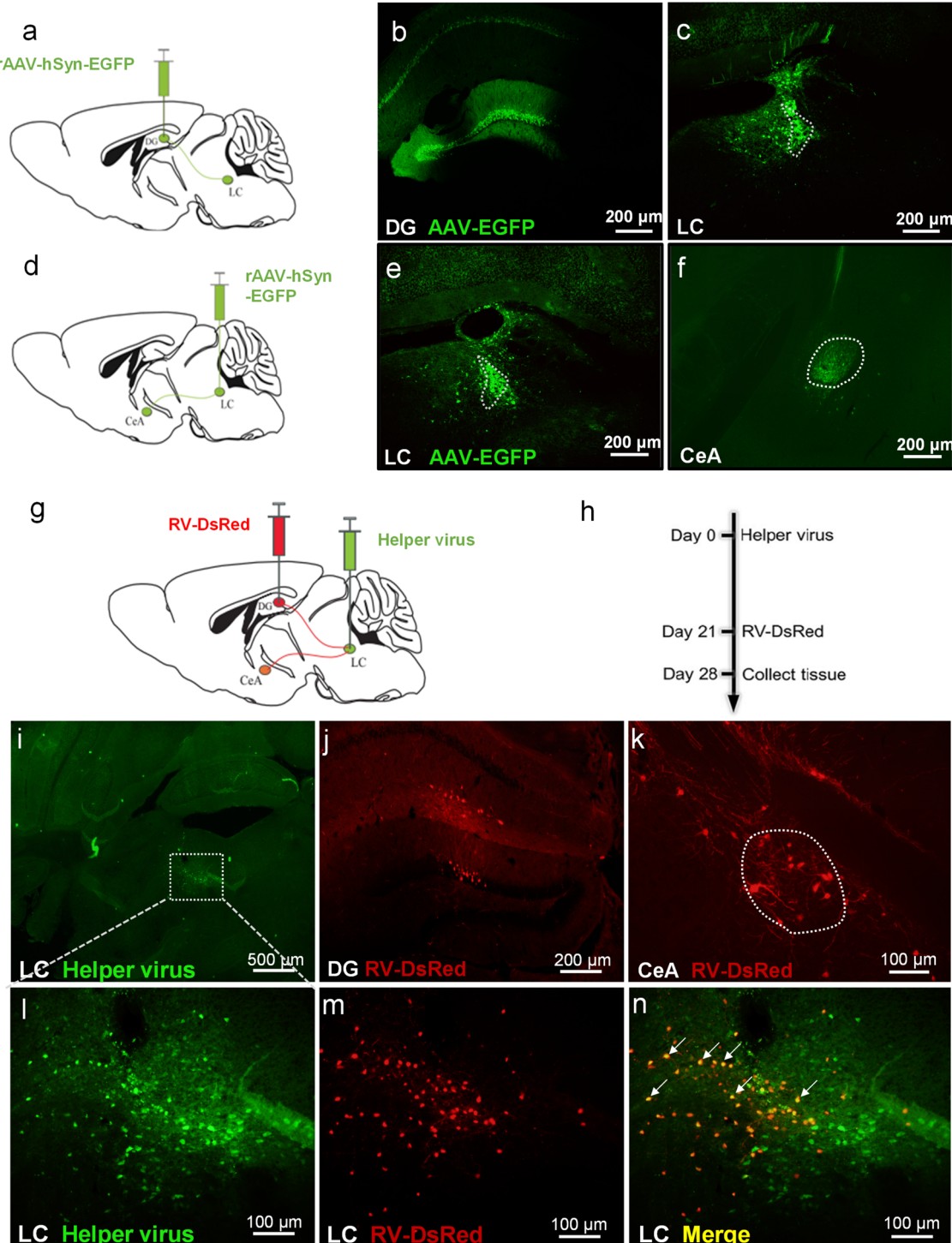

**Fig. 3 The LC neurons receive inputs from the CeA and send projections to the DG. a** Schematic illustration of injections of the AAV-hSyn-EGFP virus into the DG. **b** The expression of EGFP in the DG after intra-DG virus injection. **c** Retrograde labeling of LC-DG projection neurons after intra-DG injection of AAV-hSyn-EGFP virus. **d** Schematic illustration of intra-LC injection of the AAV-hSyn-EGFP virus. **e** The expression of EGFP in the LC after virus injection. **f** Retrograde labeling of CeA-LC projection neurons after intra-LC injection of the AAV-hSyn-EGFP virus. **g, h** The diagram of virus injection and experimental protocol for demonstration of the three-level projection. The helper virus was injected into the LC. Three weeks later, the RV-DsRed virus was injected into the DG. **i, l** The restricted expression of EGFP in the LC after intra-LC injection of the helper virus. **j** The expression of DsRed in the DG after injection of the RV-DsRed virus. **m** Retrograde labeling of the LC-DG projection neurons after intra-DG injection of the RV-DsRed virus. **k** Retrograde expression of rabies virus in the CeA. **n** The merged image of l and m. White arrows depict double-labeled neurons which were infected by both the helper virus and rabies virus.

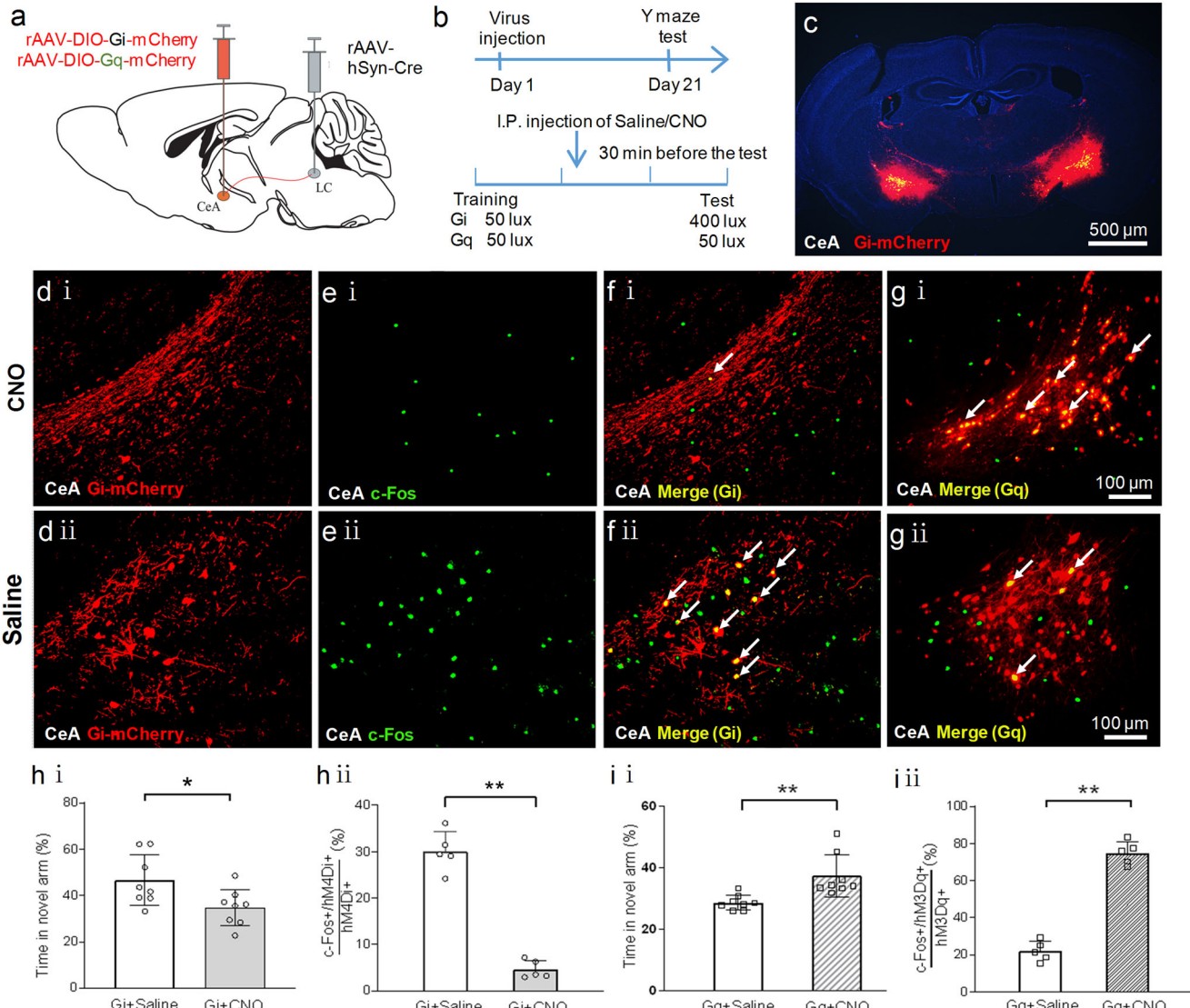

**Fig. 4 Activation of the CeA-LC circuit plays a vital role in the enhanced spatial memory retrieval induced by 400 lux light exposure. a** Schematic illustration of injecting different chemogenetic viruses into the CeA, and the injection of the rAAV-hSyn-Cre virus into the LC. **b** Experimental protocol and timeline. **c** The expression of mCherry in the CeA after injection of the rAAV-DIO-Gi-mCherry virus into bilateral CeA. **d** (i), **d** (ii) The expression of mCherry in the CeA neurons in both CNO and saline groups. **e** (i), **e** (ii) Representative images of immunofluorescence staining of c-Fos in CeA neurons after intraperitoneal injection of CNO or saline. **f** (i) The merged image of **d** (i) and **e** (i). **f** (ii) The merged image of **d** (ii) and **e** (ii). Cells co-labeled by mCherry and c-Fos were indicated by the white arrows. **g** (i), **g** (ii) Representative images of c-Fos staining and expression of mCherry in the CeA after intraperitoneal injection of CNO or saline in mice that received an intra-CeA injection of the rAAV-DIO-Gq-mCherry virus and intra-LC injection of the rAAV-hSyn-Cre virus. Cells double-labeled by mCherry and c-Fos were indicated by white arrows. **h** (i) In mice that express the inhibitory Gi-coupled receptors in the CeA, CNO induced chemogenetic inhibition of CeA neurons and decreased the time spent exploring the novel arm, as compared to saline. The light treatment during the test was 400 lux. **h** (ii) CNO also reduced the number of c-Fos+ neurons in the CeA. **i** (i) In contrast, in mice that express the excitatory Gq-coupled receptors in the CeA, CNO induced chemogenetic activation of CeA neurons and increased the time spent exploring the novel arm, as compared to saline (non-parametric Mann-Whitney U test). The light treatment during the test was 50 lux. **i** (ii) CNO also increased the number of c-Fos+ neurons in the CeA. *$P < 0.05$, **$P < 0.01$ vs. indicated group, two independent sample t-test unless otherwise noted, $n = 7$–8/group for behavioral test; $n = 5$/group for c-Fos staining.

Compared to saline, CNO reduced the time spent in the novel arm during the testing period (Fig. 4h i), suggesting that chemogenetic inhibition of the CeA-LC pathway attenuated the promoted spatial memory retrieval induced by 400 lux light exposure. Moreover, CNO decreased the number of c-Fos+ CeA-LC projection neurons, as compared to saline (Fig. 4e, f, and h ii). Thus, these findings demonstrate an indispensable function of the CeA-LC circuit in the promoted spatial memory retrieval from 400 lux light exposure.

We further tested whether activation of CeA-LC is sufficient to improve spatial memory retrieval in the control group that received 50 lux light exposure. To express the excitatory chemogenetic receptors selectively in the CeA neurons that project to the LC, we injected the rAAV/hSyn-DIO-Gq-mCherry virus bilaterally into the CeA, and rAAV-hSyn-Cre virus into the LC (Fig. 4a). CNO was injected (i.p.) 30 min before the Y-maze test to activate the CeA-LC circuit. Compared to saline, CNO did increase the time spent in the novel arm during the testing period

(Fig. 4i i), suggesting that activation of the CeA-LC circuit promoted spatial memory retrieval. CNO also increased the number of c-Fos+ CeA-LC projection neurons, as compared to saline (Fig. 4g, and i ii). Collectively, these findings prove a sufficient and indispensable function of CeA-LC projection in improving spatial memory retrieval.

Using the same strategy (Supplementary Fig. 4), we further demonstrated that chemogenetic inhibition of the LC-DG circuit also hindered the promoted spatial memory retrieval induced by 400 lux light exposure (Supplementary Fig. 4h i), and reduced the expression of c-Fos in LC-DG projection neurons (Supplementary Fig. 4f, h ii). In contrast, chemogenetic activation of the LC-DG circuit promoted spatial memory retrieval during the test period (Supplementary Fig. 4i i) and increased the number of c-Fos+ LC-DG projection neurons (Supplementary Fig. 4g, i ii).

**Neuronal subtypes involved in the light-induced facilitation of spatial memory retrieval.** So far, our findings have demonstrated the sufficiency and necessity of both CeA-LC and LC-DG circuits in promoting spatial memory retrieval. Yet, the neurochemical properties of CeA, LC, and DG neurons that are involved in this process remain unclear. It was reported that anxiety-related CRH+ neurons in the CeA project to the LC, and most of the LC neurons that receive projection from the CeA are TH+[31]. To determine whether CeA-LC projection neurons are CRH+, the rAAV-CRH-EYFP virus was injected into the CeA to label CRH+ neurons (Fig. 5a, b i and b ii), and the rAAV-hSyn-mCherry virus was injected into the LC (Fig. 5a, c) to retrograde label CeA-LC projection neurons (Fig. 5d). As a result, CRH and mCherry double-stained cells would represent CRH+ CeA neurons projecting to the LC (i.e., CeA-LC projection neurons), and our findings show that a large portion (77%) of CeA-LC projection neurons were CRH+ (Fig. 5e, f).

To examine the neurochemical properties of LC neurons that receive CeA projections, we injected the anterograde rAAV-hSyn-mCherry virus into the CeA (Fig. 5g, i), which labeled the LC neurons receiving CeA inputs (Fig. 5j). In addition, the rAAV-TH-EYFP virus was injected into the LC to identify TH+ neurons (Fig. 5h i, h ii). Thus, double-stained LC cells would be TH+ neurons receiving CeA projections, and 86% of LC neurons that receive CeA projections were TH-positive (Fig. 5k, l). Using a similar approach, we further demonstrated that 77% of LC neurons that project to the DG were TH+, and 62% of DG neurons receiving LC projections were CaMKIIα+ (Supplementary Fig. 5).

**Contribution of corticotropin-releasing factor (CRF) in LC.** To identify key neurochemical mechanisms involved in the process of promoted spatial memory retrieval, we first examined changes in neurotransmitter release in the LC in vivo after 400 lux light exposure in real time. The CRH+ CeA neurons can release CRF into the LC, which may mediate stress-induced anxiety[31]. Accordingly, we injected the rAAV-hSyn-CRF virus into the LC to express a genetically encoded CRF sensor that enables rapid and specific detection of CRF[32] (Fig. 6a), and this neurotransmitter probe was successfully expressed in the LC (Fig. 6c). In the Y-maze test, dynamic changes of CRF in the LC were measured in awake, free-moving mice by optical fiber recording. Exposing these mice to 400 lux light increased the fluorescence level in the LC, indicating an increase in CRF content (Fig. 6b, d). We next tested if CRF receptor antagonists can block the effect induced by 400 lux light exposure. Since CRF mainly activates CRFR1 in the LC, antalarmin[31] (a selective CRFR1 antagonist) or ACSF (control) was infused into the bilateral LC via embedded cannulas (Fig. 6e) 10 min before the Y-maze test. In ACSF pre-

treated mice, 400 lux light exposure promoted spatial memory retrieval during the testing period, as compared to 50 lux control. However, this effect was hindered by antalarmin pre-treatment (Fig. 6f). Collectively, these findings suggest that CRF is released into the LC during 400 lux light exposure, and contributes to the promoted spatial memory retrieval.

**Involvement of norepinephrine (NE) but not dopamine (DA) in the DG.** Since TH+ LC neurons that project to the DG may release NE and DA, we examined whether these neurotransmitters in the DG affect memory retrieval in current study. The NE and DA sensor virus, which are robust tools for rapid and specific monitoring of in vivo NE or DA transmission[12,33], were injected into the DG and an optical fiber was implanted over the injection site (Supplementary Fig. 6a, di, and dii). The 400 lux light exposure during the testing period increased the fluorescence level in the DG, indicating an increased NE release (Supplementary Fig. 6b, e). Moreover, intra-DG injection of propranolol[34] (a β-norepinephrine receptor antagonist) blocked the promoted spatial memory retrieval from 400 lux light exposure, as compared to ACSF control (Supplementary Fig. 6c, f). By contrast, intra-DG injection of SCH23390 (a D1/D5 dopamine receptors antagonist) was not effective (Supplementary Fig. 6g–l). These findings suggest that NE, but not DA, signaling in the DG contributes to the promoting effect of light exposure on spatial memory retrieval.

**Activation of the CeA-LC circuit excites CaMKIIα+ neurons.** Lastly, we determined if the functional CeA-LC-DG circuit consists of CeA-LC projection neurons and CaMKIIα+ DG neurons. The rAAV-DIO-Gq-mCherry virus and rAAV-hSyn-Cre virus were injected into the CeA and LC, respectively. This treatment led to the expression of excitatory chemogenetic receptors selectively in CeA neurons which project to the LC (Fig. 7a, b, and c). In addition, the AAV-CaMKIIα-GCaMP6m virus was injected into the DG, and an optical fiber was embedded for imaging calcium responses in CaMKIIα+ DG neurons (Fig. 7a, d). In this way, activation of CeA-LC projection neurons would increase the fluorescence level in the DG, and suggest the existence of the CeA-LC-DG functional circuitry. Indeed, intra-CeA administration of CNO, but not ACSF, significantly increased calcium signals in the DG, which indicates an increased activation of CaMKIIα+ DG neurons in vivo (Fig. 7e).

## Discussion

Previous studies of the impact of light treatment on learning and memory have yielded mixed results[5–8], which may be partially attributed to differences in the paradigms of behavioral test and light intervention. Bright light can be an aversive stimulus to nocturnal mice, rendering stress and anxiety[13,14], which can impact learning and memory processes[15,16]. To discern changes in spatial memory and anxiety level induced by light exposure, we adopted the Y-maze test which was shown to cause minimal stress[18], thus avoiding the interferences of stress and anxiety from paradigm itself.

Current findings show that light exposure increased the level of stress and anxiety in mice, as indicated by decreased exploration time in the center zone in the open field test, and decreased time in the open arm in the elevated plus-maze test. Yet, at a moderate intensity of 400 lux, light enhanced spatial memory retrieval in the Y-maze test. In contrast, the higher intensities (800 lux, 1200 lux) of light induced the opposite effect. These findings suggest that light exposure at a mild/moderate intensity (e.g., 400 lux) may induce an "optimal" level of stress which may be beneficial to learning and memory. It is plausible that a low level

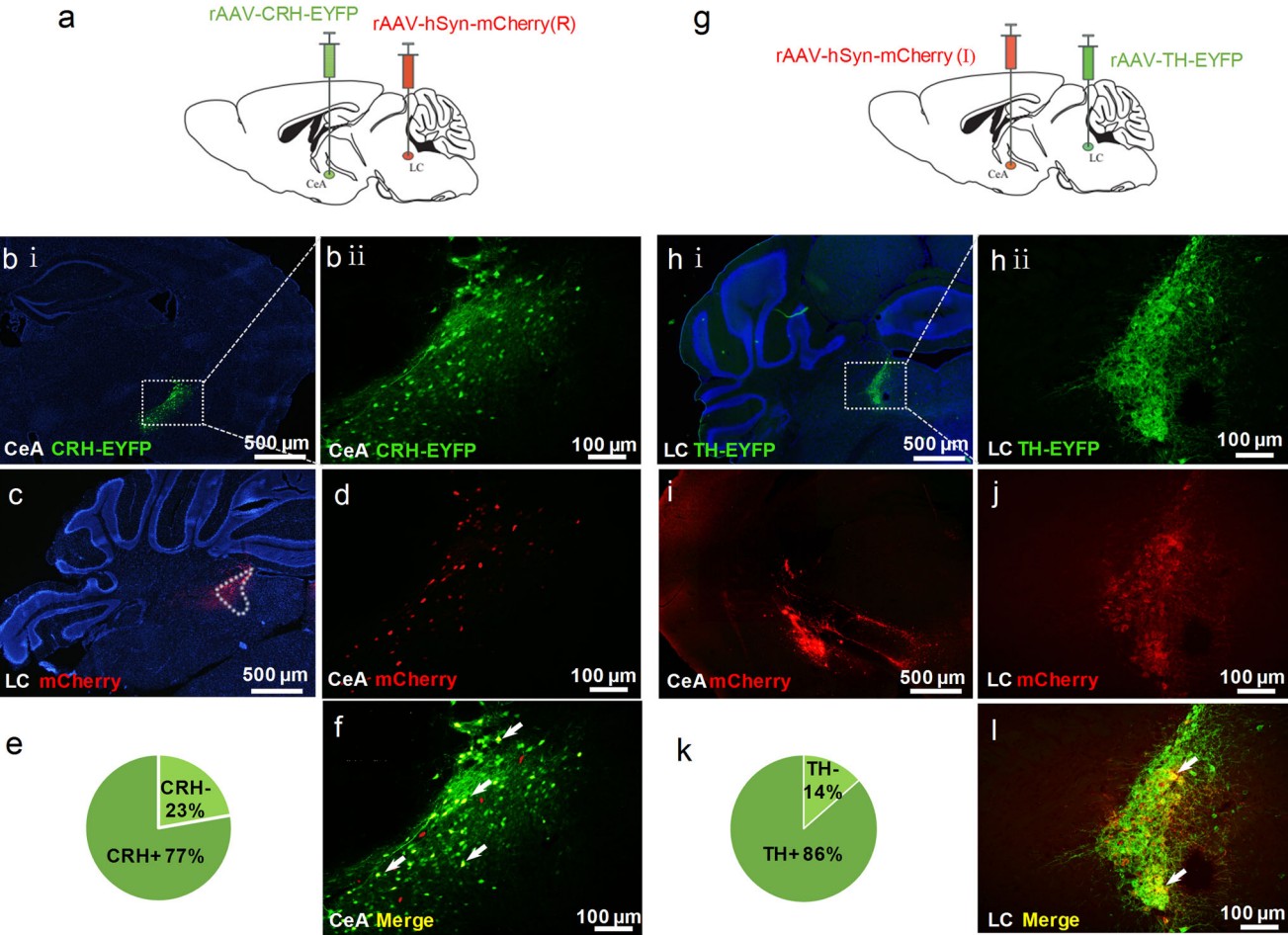

**Fig. 5 CRH + neurons in the CeA predominantly project to TH + neurons in the LC. a** Schematic illustration of intra-CeA injection of the rAAV-CRH-EYFP virus and intra-LC injection of the rAAV-hSyn-mCherry virus. **b** (i), **b** (ii) Intra-CeA injection of the AAV-CRH-EYFP virus induced restricted expression of EYFP (green) in the CeA. EYFP has selectively expressed in CRH + neurons. Intra-LC injection of the AAV-hSyn-mCherry virus induced expression of mCherry in the LC (**c**), and retrogradely labeled CeA-LC projection neurons in the CeA (**d**). **e** The proportion of CRH + (77%) in mCherry-retrogradely labeled CeA-LC projection neurons. **f** EYFP (green) and mCherry (red) double-labeled neurons in the CeA were indicated by white arrows and were CRH + CeA neurons projecting to the LC. **g** Schematic illustration of intra-CeA injection of the rAAV-hSyn-mCherry virus and intra-LC injection of the rAAV-TH-EYFP virus. **h** (i)-**h** (ii) Intra-LC injection of the AAV-TH-EYFP virus-induced restricted expression of EYFP (green) in the LC. EYFP has selectively expressed in TH + neurons. Intra-CeA injection of the AAV-hSyn-mCherry virus-induced expression of mCherry in the CeA (**i**), and anterogradely labeled neurons in the LC (**j**). **k** The proportion of TH + (86%) cells in mCherry-anterogradely labeled LC neurons, which receive projections from the CeA. **l** EYFP (green) and mCherry (red) double-labeled neurons in the LC were indicated by white arrows and were TH + LC neurons receiving projections from the CeA.

of stress may rather increase animals' vigilance and attention to environmental cues, and thus facilitate the retrieving of stored memory. However, high-intensity light exposure may cause overly stress and excessive anxiety which are aversive and harmful, and thus decreased exploration and memory retrieval. Current findings also demonstrate an inverted U-shape relationship between anxiety level and memory, which is in line with the dose-response function proposed by Yerkes-Dodson[35].

Our study further unraveled an important neural circuit underlying the beneficial effects of moderate light exposure (Fig. 8). By conducting c-Fos mapping and in vivo calcium imaging experiments, we demonstrated increased neuronal activations in three brain regions (CeA, LC, DG) during the 400 lux light exposure. These brain regions were also activated by light stimulation in humans, as shown in the brain fMRI[25,26]. Mounting evidence has shown the anatomical and functional connections among these brain regions. For instance, a previous study demonstrated axonal projections from LC neurons to the dorsal hippocampus, and activation of this pathway during the

memory encoding period improved spatial memory[34]. A connection between the CeA and LC has also been shown[28,29]. Here, we also confirmed the CeA-LC and LC-DG projections by using the virus tracing technique. Importantly, we demonstrated, for the first time, the synaptic connections from the CeA to LC, and then from the LC to DG. In this circuit, the activation of CeA neurons was suggested to be associated with anxiety, the activation of LC may increase alertness, and DG neurons are important to spatial memory[36]. Using chemogenetic approaches, we further demonstrated that both CeA-LC and LC-DG projections play sufficient and indispensable functional roles in the enhancement of spatial memory retrieval by moderate light exposure. Current findings support the notion that inputs from the CeA regulate LC neuron activity, and increase the alertness state which improves task performance[31]. Collectively, these findings suggest that a mild level of anxiety induced by moderate white light exposure may increase alertness and attention, thus increasing sensitivity to external cues and facilitating memory retrieval. It remains to be determined how high-intensity light exposure reduced memory

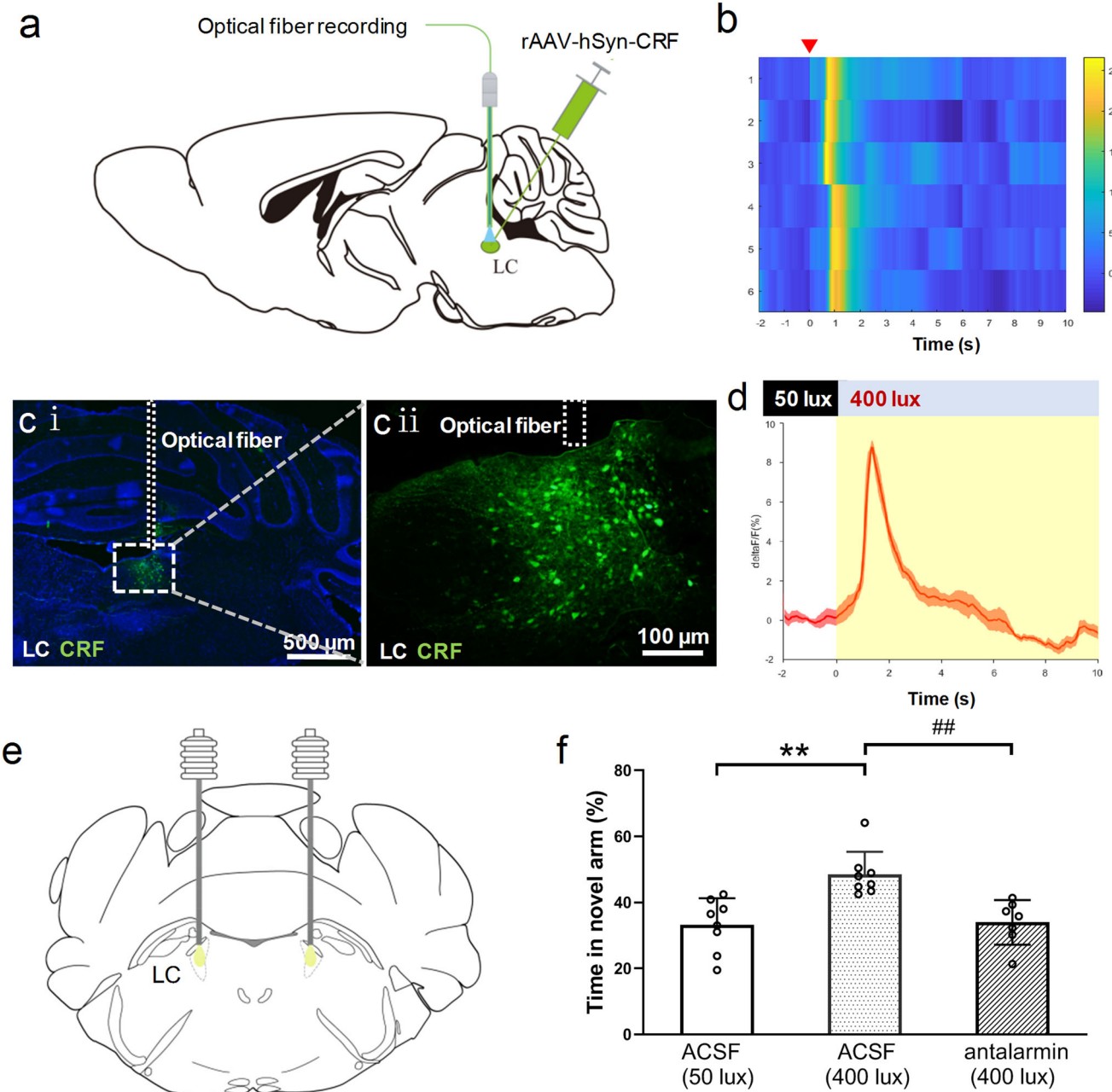

**Fig. 6 CRF was released from CeA-LC projection neurons and activated LC neurons to promote spatial memory retrieval. a** Schematic illustration of intra-LC injection of the AAV-hSyn-CRF virus, and the implantation of optical fibers above the injection site. **b** The heat map of CRF signals in the LC after 400 lux light exposure in six mice. **c** (i) **c** (ii) Intra-LC injection of the AAV-hSyn-CRF virus-induced restricted expression of CRF in the LC. The dotted lines represent the embedded optical fiber. **d** The averaged CRF signal induced by 400 lux light exposure ($n = 6$). **e** The schematic diagram of bilateral cannula implantations in the LC. **f** Quantification of time spent in the novel arm in the Y-maze test. ACSF (50 lux): 50 lux light exposure was adopted both during the training and testing periods, and ACSF was infused into the LC 10 min before the testing period; ACSF (400 lux): 50 lux light exposure was adopted during the training period and 400 lux light was used during testing, and ACSF was infused into the LC 10 min before the testing period. Antalarmin (400 lux): 50 lux light exposure was adopted during the training period and 400 lux light was applied during testing. The CRFR1 antagonist antalarmin was infused into the LC 10 min before the testing period. Data are expressed as mean ± SD. **$P < 0.01$, ##$P < 0.01$ vs. indicated group, non-parametric Kruskal-Wallis H Test, $n = 7$–8/group.

retrieval capacity. Long-term potentiation and longer-term depression, respectively, are closely related to memory enforcement and impairment. Characterizing how low and high intensities of light may differentially affect neuronal plasticity in the CeA-LC-DG circuit may provide further insights.

The brain nucleus is composed of functional distinct subtypes of neurons playing different roles in the execution of specific neural functions[37,38]. CRH + neurons are widely distributed in

the brain and regulate both emotional and cognitive processes by releasing CRF[39]. As shown in Fig. 8, our findings suggest that CeA-LC projection neurons may release CRF into the LC during 400 lux light exposure. CRF acts on the CRFR1 receptor on LC neurons and increases their activities. Both NE and DA can be released into the DG from the axonal terminals of LC-DG projection neurons[34,40–43], though our findings suggest that NE but not DA released by LC-DG projection neurons is necessary in the

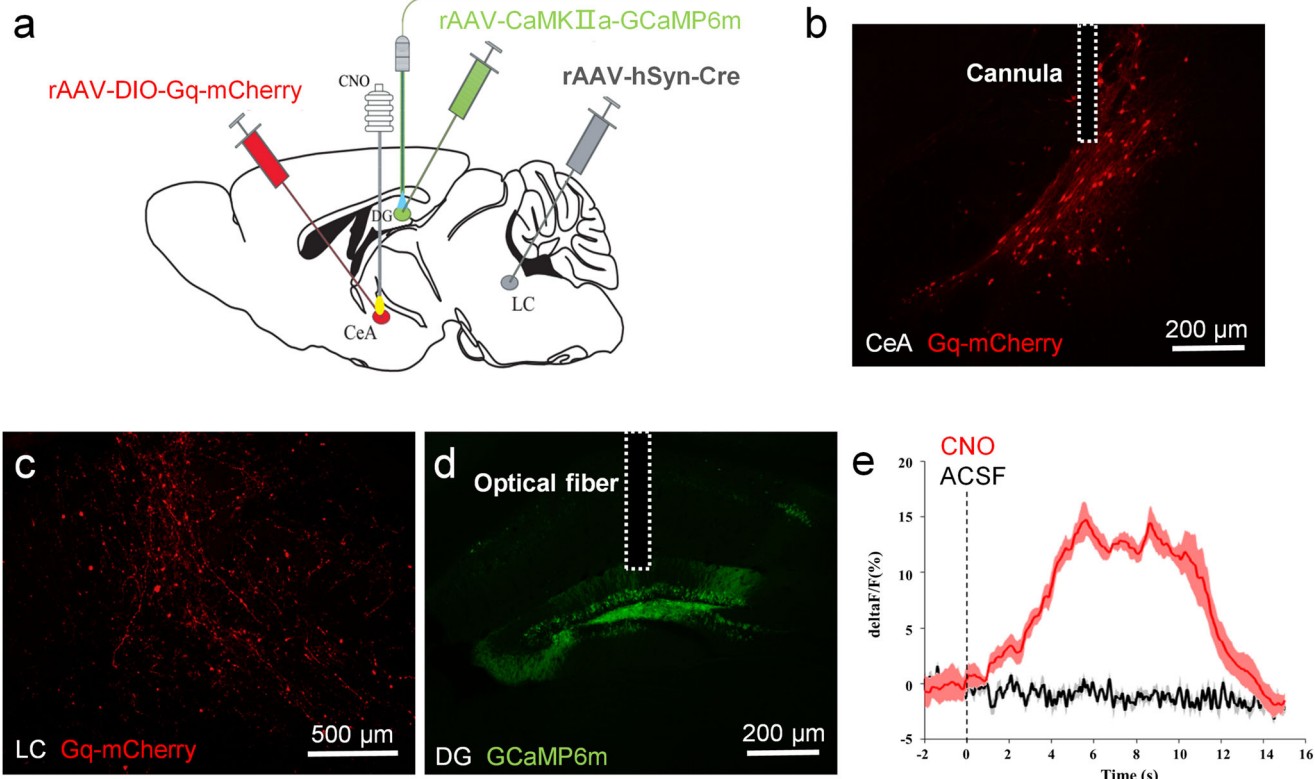

**Fig. 7 Chemogenetic activation of the CeA neurons increased the activity of CaMKIIα + neurons in the DG. a** Schematic illustration of the experimental setup for the functional examination of the CeA-LC-DG circuit. The AAV-DIO-Gq-mCherry virus was injected into the CeA, and a cannula for intra-CeA injection of CNO or saline was embedded above the injection site. The retrograde virus AAV-hSyn-Cre was injected into the LC. The AAV-CaMKIIα-GCaMP6m virus was injected into the DG, and an optical fiber was embedded above the injection site for calcium imaging. **b** Intra-CeA injection of the AAV-DIO-Gq-mCherry virus induced the expression of Gq-mCherry (red) in the CeA. The dotted lines represent the embedded cannula. **c** Gq-mCherry (red) is also expressed at the terminals of CeA-LC projection neurons in the LC. **d** Intra-DG injection of the AAV-CaMKIIα-GCaMP6m virus-induced restricted expression of GCaMP6m in CaMKIIα + neurons in the DG. The dotted lines represent the embedded optical fiber. **e** The calcium signal in the DG was increased after an intra-CeA injection of CNO, but not ACSF ($n = 5$).

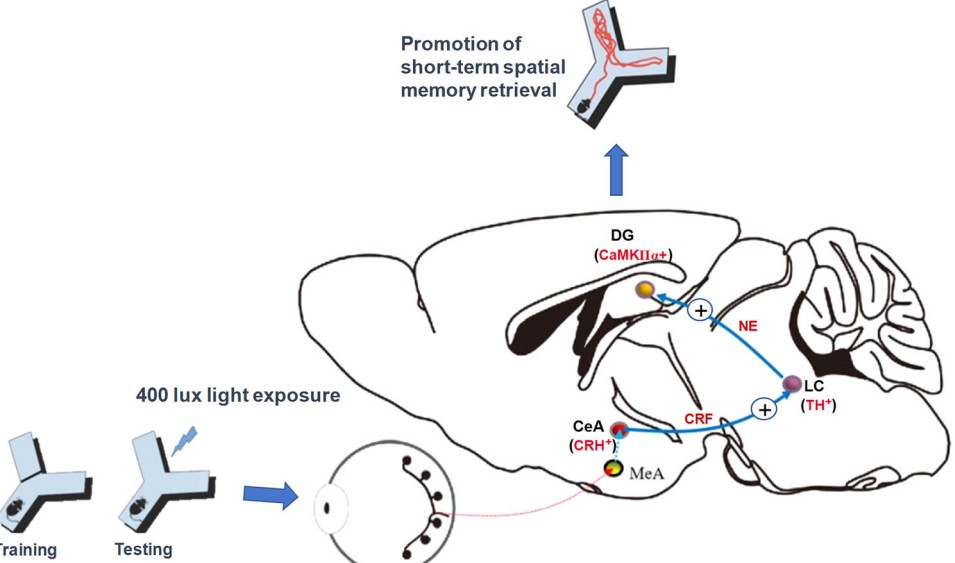

**Fig. 8 Light exposure of 400 lux could promote memory retrieval by activating CeA-LC-DG circuit.** In this process, light increased the activity of CeA - CRH+ neurons which release CRF and activate LC - TH+ neurons. TH+ neurons in LC release NE and increase the activity of CaMKIIα + neurons in DG through LC-DG circuit, thus promoting short-term spatial memory retrieval.

enhancement effect after 400 lux light exposure. The downstream DG neurons that receive projections from LC neurons were mostly CaMKIIα+. Importantly, the chemogenetic activation of the CeA-LC circuit increased the activity of CaMKIIα + neurons in the DG, thus proving a functional CeA-LC-DG trisynaptic circuit. Furthermore, we found that neurons in the MeA receive inputs from the retina and send projections to the CeA, indicating that the light information received by the retina is mediated by MeA to further activate the CeA-LC-DG circuit.

Light is detected by the retina and the information is transmitted to the brain. Our viral tracing study did not find direct anatomical connections between the retina and the CeA. Based on the broad projections of ipRGCs into the brain[44] and the ability of CeA to receive projections from multiple brain regions, light stimulation may activate CeA neurons through an indirect mechanism. For example, neurons in the MeA receive inputs from the retina and may send projections to the CeA as shown in the current study.

To our knowledge, enhancement of spatial memory retrieval by moderate light exposure has not been reported before, nor have the underlying neural mechanisms. Our findings unraveled a functional CeA-LC-DG circuit that mediates this acute beneficial effect and further identified the neuron subtype and major neurochemical mechanisms involved. Specifically, 400 lux light exposure during the testing period of the Y-maze test activated the CeA-LC-DG circuit to promote spatial memory retrieval. During this process, CRF is released from the axon terminals of CRH + CeA neurons and activates CRFR1 receptors in LC neurons; The LC neurons then release NE from their axon terminals ending in the DG, and NE binds to β-type noradrenaline receptors to activate CaMKIIα + DG neurons. Phototherapy has been tested clinically to improve cognitive decline and related disorders[9,45]. The elucidation of the beneficial effects of moderate light on cognition and potential underlying mechanism will provide a theoretical basis for the formulation of new phototherapies. Moreover, the identification of the underlying neural circuit and the neurochemical mechanisms may provide new targets for regulating anxiety-related memory function, and hence may have important clinical implications.

There are some limitations in this study. First, the transmission pathway from the retina to CeA is not clear. Among three types of photoreceptors in the retina, ipRGC was considered to be the primary mediator of light regulation for non-visual functions such as memory. Nevertheless, future studies are needed to examine whether cones and rods also play a role. Second, our findings suggest that MeA may be important for transmitting information from the retina to CeA, but other brain nuclei that may also be involved in this process remain to be determined. Third, mice are known to be nocturnal, and some findings from animal behavior tests may not be applicable to humans. Therefore, future studies in diurnal animals and advanced non-human primates will provide further evidence for the translational potential.

Our study thus demonstrated a specific light scheme that can promote spatial memory without excessive stress, and unraveled the underlying CeA-LC-DG circuit and associated neurochemical mechanisms.

## Methods

**Animals**. Adult (20–23 g) male C57BL/6J mice were group-housed (5 mice per cage), given food pellets and water ad libitum, and maintained under constant environmental conditions (temperature 23 ± 2 ℃, humidity 55 ± 5%, and 12:12 h light/dark cycle, 200 lux white ambient illumination). All animal handling procedures were conformed to the guidelines for Laboratory Animal Research of Fourth Military Medical University. The Institutional Animal Care and Use Committee of Fourth Military Medical University approved the animal protocols and experiments.

**Illumination parameters**. Different intensities (50, 100, 200, 400, 800, and 1200 lux) of white light were used in the behavioral experiment. Full-spectrum light was supplied by the panel lamp (composed of cool LED lights) placed above the behavior testing apparatus. The brightness at the bottom surface of the behavior equipment was measured by the illuminometer and kept consistent. The light intensity was determined by averaging the measurements from the center and four sides of the bottom surface.

**Behavioral paradigms**. All behavioral tests were conducted during the light phase (9 A.M.–1 P.M.). Operators were blinded to the experimental group during the testing and scoring.

*The Y maze test*. The Y maze test was conducted largely according to the method described previously. There are two stages: training and testing. During the training period, the animals were allowed to move freely in two arms of the maze for 10 min with the third arm blocked. At the end of the training period, the animal was returned to its home cage. The testing was conducted after 1 h, during which the blocked arms were opened and the animal could move freely to any of the three arms for 5 min in the maze. The number of entries into the new arm (the arm blocked during the training period) and the total time spent in the new arm was measured and compared between groups (Supermaze software).

*Novel object location test (NOL)*. The mice were habituated continuously in the experimental room for 3 days before the training phase, during which the mice were allowed to explore the open field (30 cm length × 30 cm width × 38 cm height) freely under standard light (50 lux), 10 min/d. The 50 lux light intensity was used during the training period, and two identical objects were placed at the two diagonal corners of the arena 10 cm from the wall. Mice were allowed to explore the object for 10 min, only once training. The test was carried out 1 h after training. Different light intensities (50, 100, 200, 400, 800, 1200 lux) were adapted during the training period. During the testing phase, one of the objects was moved to a position parallel to the other side of the wall, and the percentage of time the mice spent exploring the new location within 10 min was calculated to evaluate the mice's new object location memory. All the objects were screened in advance, and the mice had no obvious preference for these objects. After each test, the open field and objects should be sprayed with 70% ethanol and wiped clean with paper towels.

*Open field test (OFT)*. Motor activity was measured in an open field test box (30 cm length × 30 cm width × 38 cm height). The mice were placed in the middle of a plastic box and allowed to explore in the arena for 10 min, during which the light intensity of different groups of mice varied (50, 100, 200, 400, 800, 1200 lux). The movements of mice were recorded by the camera at the top of the box. The time spent in the center was measured (Supermaze software). The box was wiped clean with a paper towel soaked in 70% ethanol and dried thoroughly after each test session.

*Elevated plus-maze test (EPM)*. The apparatus for the EPM is 40 cm above the ground and consists of two opposing open arms (28 cm length, 5.8 cm width) and two closed arms (28 cm length, 5.8 cm width), which were connected by a central zone (5.8 cm length, 5.8 cm width). The whole apparatus was elevated 40 cm above the floor. The mouse was placed in the center zone facing toward one open arm and was allowed to freely explore the arena with various light intensity (50, 100, 200, 400, 800, 1200 lux) during a 5 min test session. Time spent in the open arm during the 5 min was measured (Supermaze software). The box was wiped clean with a paper towel soaked in 50% ethanol and dried thoroughly after each test session.

**c-Fos mapping**. To quantify the influence of 400 lux light exposure in the testing period on c-Fos expression, mice (n = 10) were separated into two groups in the Y maze test: 1) Co, 5 mice were trained under 50 lux light and also tested under 50 lux light; 2) LT, 5 mice were trained under 50 lux light and tested under 400 lux light. Mice were anesthetized (Avertin, 13 mL/g, i.p.) one and half an hour after the test and perfused intracardially with 0.9% saline followed by 4% paraformaldehyde in PBS. The brain was removed and then postfixed overnight in 4% paraformaldehyde, after which the brain was transferred into 30% sucrose. Frozen brain sections were prepared by using a cryostat (CM1900, Leica Microsystems, Bannockburn, IL).

For c-Fos labeling, brain sections were placed in 0.1 M PBS containing 10% normal goat serum (Vector Laboratories, Burlingame, CA) and 0.3% Triton X-100 (T8787, Sigma-Aldrich, St Louis, MO) for 1 h before incubation in the primary antibody against c-Fos (rabbit, 1:800, 2250, Cell Signaling Technology, 18 h at room temperature). Then the sections were incubated with a specific secondary antibody (Alexa Fluor 488, goat anti-rabbit, 1:400, 111-545-003, Jackson ImmunoResearch) at a dilution of 1:400 for 4 h at room temperature. Finally, all sections were rinsed in 0.1 M PBS and coverslipped in an anti-fading aqueous mounting medium with DAPI (EMS, Hatfield, PA). The expression of c-Fos in the whole brain was observed under a fluorescence microscope. The number of c-Fos+ neurons in each brain region was counted with Image J, and the area was measured to calculate the number of c-Fos+ neurons per square millimeter. The average

number of c-Fos+ cells in each brain region was calculated as the total number of c-Fos+ cells counted in 5 mice divided by the number of animals.

**Virus injection**. Mice were anesthetized with pentobarbital (i.p., 80 mg/kg), placed in a stereotaxic apparatus (RWD, China), and kept warm (37 °C) with an electric heating pad (BrainKing Biotech, China). Erythromycin eye ointment was applied to prevent corneal drying. During surgery and virus injection, mice were kept anesthetized with 1–1.5% isoflurane mixed with pure oxygen at a 1 L/min flow rate. After the scalp was cut open, 3% hydrogen peroxide was applied to remove the fascia from the skull surface. The head was adjusted to the horizontal position according to the bregma and lambda points. The skull over the targeted brain region was thinned with a dental drill (OmniDrill35, WPI, Sarasota, FL) and carefully removed. A small craniotomy hole with a diameter of 300 to 500 µm was opened for virus injection.

The virus injections were performed at 40–60 nL/min for a total volume of 200–300 nL, using a micropipette connected to the Nanoliter Injector (NANOLITER 2010, WPI, Sarasota, FL) with a controller (Micro4, WPI, Sarasota, FL). After the injection, the injector was kept in place for 10 min and then extracted slowly to minimize solution leakage in the injection track. Finally, the wound was closed except if implantation of optical fiber or cannula is needed. Experiments were performed 2-3 weeks after virus injection.

Coordinates for intra-CeA injection (200 nL/injection) were: anterior-posterior (AP), −1.40 mm; medial-lateral (ML), ± 2.70 mm; dorsal-ventral (DV), −4.75 mm. Intra-LC injection (150 nL/injection) coordinates were: AP, −5.50 mm; ML, ± 0.85 mm; DV, −4.30 mm. Intra-DG injection (200 nL/injection) coordinates were: AP, -2.06 mm; ML, ± 1.30 mm, DV, −1.64 mm. When performing fiber photometry and neural tracing, unilateral injections were adopted. When performing chemogenetic experiments, bilateral injections were used.

To infect CeA-LC projection neurons, the rAAV2-Retro-Cre virus was injected into the LC (virus titers: $5.38 \times 10^{12}$ vg/mL), the AAV2/9-DIO-hM3Dq-mCherry (virus titers: $5.21 \times 10^{12}$ vg/mL) or AAV2/9-DIO-hM4Di-mCherry (virus titers: $5.27 \times 10^{12}$ vg/mL) virus was injected into the CeA. To infect LC-DG projection neurons, the rAAV2/2-Retro-Cre virus was injected into the DG (virus titers: $5.38 \times 10^{12}$ vg/mL) and the AAV2/9-DIO-hM3Dq-mCherry (virus titers: $5.21 \times 10^{12}$ vg/mL) or AAV2/9-DIO-hM4Di-mCherry (virus titers: $5.27 \times 10^{12}$ vg/mL) virus was injected into the LC.

To infect CRH + CeA neurons with EYFP, the AAV2/9-CRH-EYFP virus was injected into the CeA (virus titers: $4.92 \times 10^{12}$ vg/mL). In addition, the AAV2/R-hSyn-mCherry virus was injected into the LC (virus titers: $5.27 \times 10^{12}$ vg/mL). To infect TH + LC neurons with EYFP, the AAV2/9-TH-EYFP virus was injected into the LC(virus titers: $4.86 \times 10^{12}$ vg/mL). Moreover, the AAV2/1-hSyn-mCherry virus was injected into the CeA (virus titers: $5.38 \times 10^{12}$ vg/mL), or AAV2/R-hSyn-mCherry virus was injected into the DG (virus titers: $5.27 \times 10^{12}$ vg/mL). To infect CaMKIIα + DG neurons with mCherry, the AAV2/9-CaMKIIα-mCherry virus was injected into the DG (virus titers: $5.17 \times 10^{12}$ vg/mL), and the AAV2/1-hSyn-EYFP virus was injected into the LC (virus titers: $5.81 \times 10^{12}$ vg/mL).

For di-synaptic tracing the CeA-LC-DG pathway, the AAV2/9-hSyn-EGFP-2a-TVA-2a-RVG (100 nL, virus titers: $6.71 \times 10^{12}$ vg/mL) virus was injected into the LC. Twenty-one days later, the RV-ENVA-ΔG-dsRed (200 nL, virus titers: $2.00 \times 10^8$ IFU/mL) virus was injected into the DG. For in vivo calcium imaging, the AAV2/9-hSyn-GCaMp6m (virus titers: $2.32 \times 10^{12}$ vg/mL) or AAV2/9-CaMKIIα-GCaMp6m (virus titers: $2.73 \times 10^{12}$ vg/mL) virus was injected into the corresponding brain regions. For neurotransmitter detection, the AAV2/9-hsyn-CRF (virus titers: $8.07 \times 10^{12}$ vg/mL)/AAV2/9-hsyn-NE1h (virus titers: $3.07 \times 10^{12}$ vg/mL)/AAV2/9-hsyn-DA1h (virus titers: $3.11 \times 10^{12}$ vg/mL) virus was injected into the corresponding brain regions (LC/DG/DG).

**Verification of injection site**. After transcardial perfusion with 0.9% saline followed by 4% paraformaldehyde in 0.1 M PBS, the brain was removed and post-fixed with 4% paraformaldehyde overnight at 4 °C, and then transferred into 30% sucrose. Brain sections were made with a cryostat (CM1900, Leica Microsystems, Bannockburn, IL). A series of 40 µm sections were collected for verification of injection sites. Tissue sections were examined under epifluorescence using a Zeiss Axioimager Z2 microscope.

**Optical fiber implantation and in vivo optical recording**. The stereotactic brain surgery procedure was the same as described above. When implanting the optical fiber (200 µm OD, 0.37 numerical aperture, Newdoon, China), the insertion speed was slow, and the tip was placed at ~0.15 mm above the targeted regions. The optical fiber was supported by two skull-penetrating M1 screws and fixed on the surface of the skull with dental cement. When the dental cement was thoroughly dried, the mice were taken off the stereoscopic positioning instrument and transferred to an electric blanket to keep warm (37 °C) till they awoke. Then, the mice were put back in their home cage.

Fiber recording was conducted according to procedures described in previous studies. The fiber photometry system was used to record fluorescence signals associated with neuronal activity. The excitation laser from a light-emitting diode at 488 nm was reflected by a dichroic mirror (MD498; Thorlabs), focused through a 10 x objective lens (NA = 0.3; Olympus) and then connected to an optical commutator (Doric Lenses, R3896, Hamamatsu, Japan). An optical fiber (O.D. = 230 mm, NA = 0.37, 2 m) was used to guide the light between the fiber photometry system and the implanted optical fiber. The laser power at the tip of the optical fiber was measured with a power meter and adjusted to 20 µW to minimize the photobleaching effect.

The GCaMP6m fluorescence signals were bandpass filtered (MF525-39, Thorlabs) and collected by a photomultiplier tube (R3896, Hamamatsu). The current output from the photomultiplier tube was converted to voltage signals by an amplifier (C7319, Hamamatsu), and then filtered through a low-pass filter (40 Hz cut-off; Brownlee 440). The analog voltage signals were converted to digital signals with a Power 1401 digitizer at 500 Hz and then recorded by Spike2 software (CED, Cambridge, UK).

With the methods described above, we recorded the calcium signals and neurotransmitter signals (CRF, NE, and DA) induced by 400 lux light exposure during the testing period of the Y maze test. The recordings were made after the animals had recovered from surgery for at least 2 weeks. During optical recording, mice were allowed to move freely in the home cage for at least 5 min and then trained in the Y maze. During the training period and before the testing period, the optical fiber was connected, but no fluorescence signal was collected and no excitation light was applied. Fluorescence signal in the target brain region was collected at the beginning of the testing period, and 50 lux light was applied during the first 3 min of the testing period and then the brightness was increased to 400 lux.

The fiber photometry data were exported from Spike2 to MATLAB for further analysis. To calculate the average response, we first segmented and aligned the raw data based on the onset of events with individual trials. To normalize the data, the fluorescence change (ΔF/F) values were calculated by $(F – F0)/F0$, where $F0$ is the baseline fluorescence signal averaged over a control time window of at least 20 s, and F is the real-time fluorescence signal. The percentages of ΔF/F values were presented with heatmaps or as average plots with a shaded area indicating the SEM.

**Cannula implantation and drug microinjection**. The surgical procedure was the same as described above. A stainless-steel guide cannula (26 gauge, 3.5 mm, RWD Life Science) was implanted 0.5 mm above the target nucleus and fixed to the skull with adhesive luting cement and acrylic dental cement. For intra-LC infusion, double guide cannula (26 gauge, Plastics One, RWD Life Science) were bilaterally implanted using the following coordinates: −5.50 mm AP, ± 0.85 mm ML, − 4.30 mm DV. For intra-DG infusion, double guide cannula (26 gauge, Plastics One, RWD Life Science) were bilaterally implanted using coordinates: −1.7 mm AP, ± 1.0 mm ML, − 2.1 mm DV. For intra-CeA infusion, a single guide cannula (26 gauge, Plastics One, RWD Life Science) was unilaterally implanted using coordinates: −1.40 mm AP, ± 2.70 mm ML, − 4.75 mm DV. Following surgery, a stainless-steel obturator was inserted into the guide cannula to avoid obstruction. Mice were given 1–2 weeks to recover after surgery.

Before the drug infusion, mice were briefly head-restrained while the stainless-steel obturator was removed and an injection tube (30 gauge, 4.0 mm, RWD Life Science) was inserted into the guide cannula. The injection tube was designed to protrude 0.5 mm from the tip of the catheter, thus reaching the target nucleus. The drug or vehicle was slowly infused at a flow rate of 80 nL per min to a total volume of 200 nL (Harvard Apparatus, Holliston, MA). Following injection, the injection cannula was left in place for 3 min to reduce backflow. The stainless-steel obturator was subsequently reinserted into the guide cannula.

The following drugs were used: antalarmin hydrochloride (a CRFR1 antagonist, 0.1 mg/mL) in the LC, propranolol hydrochloride (a beta-AR antagonist, 0.4 mg/mL) and SCH23390 (a D1/D5 antagonist, 1.13 mg/mL) in the DG, and CNO (0.05 mg/mL) in the CeA. Drug or vehicle (ACSF) was infused 30 min before the testing period in the Y maze test to assess the effect. CNO was delivered to modulate the CeA-LC circuit when calcium imaging was performed in the DG. Animals that received failed infusions (as determined by lack of movement of a fluid indicator in the clear tubing) were eliminated from the study before decoding control versus experimental groups.

**Chemogenetic manipulation**. In the behavioral test, CNO (2 mg/kg, C0832, Sigma-Aldrich, St Louis, MO) was administrated by an intraperitoneal injection 30 min before the testing period of the Y maze test. After the injection, the mice were returned to their home cage.

**Quantification of c-Fos expression after chemogenetic manipulation**. To quantify the effect of chemogenetic activation of CeA-LC/ LC-DG -projection neurons on c-Fos expression, mice received an intra-LC/intra-DG injection of the rAAV2/R-hSyn-Cre virus and intra-CeA/intra-LC injection of the AAV2/9-DIO-hM3Dq-mCherry virus, and were divided into 2 groups: 1) 50 lux-Saline, 5 mice conducted the Y maze training (under 50 lux light) and 30 min later received i.p. injection of saline, followed by a 30 min interval and then performed the testing under 50 lux light; 2) 50 lux-CNO, 5 mice conducted the Y maze training and 30 min later received i.p. injection of CNO, followed by a 30 min interval and then performed the testing under 50 lux light. One and half an hour after the test, all animals were anesthetized and perfused. The CeA/LC was subjected to immunostaining of c-Fos. In each mouse, the number of hM3Dq-labeled CeA/LC neurons and c-Fos/hM3Dq double-labeled CeA/LC neurons were counted from four

serial brain sections (40 μm/section). The percentage of c-Fos/hM3Dq double-labeled CeA/LC neurons in each group was calculated as the percentage of the total number of c-Fos/hM3Dq double-labeled CeA/LC neurons within the total number of hM3Dq-labeled CeA/LC neurons ($n = 5$ mice).

To quantify the influence of chemogenetic inhibition of CeA-LC/LC-DG projection neurons on c-Fos expression in the CeA/LC, mice ($n = 20$) received an intra-LC/intra-DG injection of the rAAV2/2-Retro-Cre virus and intra-CeA/intra-LC injection of the AAV2/9-DIO-hM4Di-mCherry virus and were divided into two groups: 1) 400 lux-Saline, 5 mice conducted the Y maze training (under 50 lux) and 30 min later received i.p. injection of saline, followed by a 30 min interval and then performed the testing (under 400 lux); 2) 400 lux-CNO, 5 mice conducted the Y maze training (under 50 lux) and 30 min later received i.p. injection of CNO, followed by a 30 min interval and then performed the testing (under 400 lux). One and half an hour after the test, all animals were anesthetized and perfused. The CeA/LC was subjected to immunostaining of c-Fos. In each mouse, the number of hM4Di-labeled CeA/LC neurons and c-Fos/hM4Di double-labeled CeA/LC neurons were counted from four consecutive brain sections (40 μm/section) across the CeA/LC. The percentage of c-Fos/hM4Di double-labeled neurons in each group was calculated as the percentage of the total number of c-Fos/hM4Di double-labeled CeA/LC neurons within the total number of hM4Di-labeled CeA/LC neurons counted in five mice.

**Quantification of neurons transfected by virus**. To quantify the percentage of CeA-LC projection neurons co-labeled with mCherry and EYFP, mice ($n = 3$) received an intra-CeA injection of the AAV2/9-CRH-EYFP virus and intra-LC injection of the AAV2/R-hSyn-mCherry virus. In each mouse, the number of mCherry-labeled neurons and mCherry/EYFP double-labeled neurons were counted in four serial brain sections (40 μm/section) across the CeA. The percentage of mCherry/EYFP double-labeled neurons was calculated as a percentage of the total number of mCherry/EYFP double-labeled neurons from the total number of mCherry-labeled neurons counted in 3 mice.

To quantify the percentage of LC neurons receiving projections from the CeA and were co-labeled with mCherry and EYFP, mice ($n = 3$) received an intra-LC injection of the AAV2/9-TH-EYFP virus and inta-CeA injection of the AAV2/1-hSyn-mCherry virus. In a similar way as described above, the percentage of mCherry/EYFP double-labeled neurons were calculated in three mice.

To quantify the percentage of LC-DG projection neurons that were co-labeled with mCherry and EYFP, mice ($n = 3$) received an intra-LC injection of the AAV2/9-TH-EYFP virus and intra-DG injection of the AAV2/R-hSyn-mCherry virus. Similarly as mentioned above, the percentage of mCherry/EYFP double-labeled neurons was calculated in three mice.

To quantify the percentage of DG neurons receiving projection from the LC and co-labeled with mCherry and EYFP, mice ($n = 3$) received an intra-LC injection of the AAV2/1-hSyn-EYFP virus and intra-DG injection of the AAV2/R-CaMKIIα-mCherry. Similarly as mentioned above, the percentage of mCherry/EYFP double-labeled neurons was calculated in three mice.

**The disynaptic connection in the CeA-LC-DG circuit**. To quantify the numbers of starter cells in the LC and RV-DsRed-labeled presynaptic neurons in the CeA, mice ($n = 3$) received an intra-LC injection of helper virus and intra-DG injection of the SAD-DG-DsRed (EnvA) virus. In each mouse, the numbers of helper virus/RV-DsRed double-labeled starter cells from four consecutive brain sections (40 μm/section) across the LC, and RV-DsRed-labeled neurons from the CeA were counted. The average numbers of starter cells in the LC and RV-DsRed-labeled presynaptic CeA neurons were calculated as the total number of starter cells and presynaptic CeA neurons counted in three animals divided by the number of animals.

**Statistics and reproducibility**. Graph Pad Prism 9.0 (GraphPad Software, USA) and MATLAB R2017b were used for statistical evaluation. Data analysis was done by experimenters blinded to experimental conditions to reduce selection and observation bias. Experimental animals were divided into different treatment groups using the random number table method, and mice in each group were tested in ascending order. All data from the different groups were verified for normality using Shapiro-Wilk before analysis to determine whether the data conforms to the Gaussian distribution and choose appropriate statistical analysis methods. Two independent sample t-test was used to compare c-Fos expression in each brain region between two groups of animals under 50 lux or 400 lux light exposure during the testing period of the Y maze. Two independent sample t-test or Mann-Whitney U test (nonparametric data) was also used to compare data between two groups after chemogenetic manipulation. One-way ANOVA with Dunnett-t multiple comparisons test was used to detect the difference in behavioral performance in the Y maze test, elevated plus-maze test and new object location test among different groups. Kruskal-Wallis H test (nonparametric data) was used to detect the difference in behavioral performance in the Y maze test, open-field test among different groups. One-way ANOVA with Dunnett-t multiple comparisons test or Kruskal-Wallis H test (nonparametric data) was also used to compare data between two groups after neurotransmitter receptor blocking experiment. All tests were two-tailed, and $P < 0.05$ was considered significant in all tests. Unless otherwise indicated, values were reported as mean ± SD.

**Reporting summary**. Further information on research design is available in the Nature Portfolio Reporting Summary linked to this article.

## Data availability

The source data underlying Fig. 1aii, bii, c-d, 2i, 4h1-iii, 6f, Supplementary Figs. S1b-e, S4hi-iii, S6f, l are provided as Supplementary Data file. xlsx. All other data are available from the corresponding authors upon reasonable request.

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

## Acknowledgements

We appreciate Professor MuMing Poo and MinMin Luo for their critical comments and constructive suggestions for revision on manuscript. We thank YuLong Li for providing AAV2/9-hsyn-CRF3.0-WPRE-pA virus. This work was supported by grants from the National Natural Science Foundation of China (51437008, 31671089).

## Author contributions

GZ.G., JL.X., J.L and Y.G. conceived the idea. JL.X., MJ.S. designed experiments and wrote the manuscript. MJ.S., ML.S., JM.Z. performed surgery. MJ.S., ML.S., RT.X., and JY.D. performed behavioral experiments and in vivo activity recordings. D.OY., JZ.D, JF.H., ZC.S., BX.W., Q.H., Y.H., and YH.L, assisted with behavioral experiments, histology, and microscopy. MJ.S., ML.S., JL.X. analyzed the data. JL.X. and Y.G. further contributed to the writing by reviewing and editing the manuscript.

## Competing interests

The authors declare no competing interests.
