## [Peer Review File · Communications Biology]

Reviewers' comments:

Reviewer #1 (Remarks to the Author):

In this manuscript, Professor Xing and colleagues have shown that short-term spatial memory in mice is improved when the animals are trained using a white light intensity of 50 lux but tested using 400 lux white light. However, those conditions induce some anxiety which is argued by the author to be the optimal amount of stress as higher light intensities during testing induce higher stress and do not enhance short-term memory retrieval in the Y maze compared to the control condition of 50 lux. In addition, the authors have tried to identify the brain circuits that are involved in this enhancement of short-term memory and they showed that CeA-LC-DG circuit is playing a crucial role using c-fos staining, calcium imaging and chemogenetics. They were also able to show that modulation of CRH+ neurons of CeA that are projecting to LC and modulation of TH+ neurons of LC that are projecting to DG impair the enhancement of the spatial memory when using 400 lux of white light during the test phase. Moreover, activation of CeA neurons projecting to LC increases the activity of CaMKIIa+ neurons in the DG. The authors, therefore, suggest a mechanism for the spatial memory retrieval at 400 lux where CRH+ neurons in the CeA activate TH+ neurons in the LC, which themselves activate CaMKIIa+ neurons in the DG.

I would like to thank the authors for sharing their work with us. The manuscript shows a huge amount of effort and dedication. Intensity of white light was shown in the past to affect learning and memory but also the level of anxiety, as mentioned by the authors in the introduction. However, very little is known about the precise neuronal mechanism of this phenomenon. This work is one of the first insights showing regions or circuits involved in this process. Overall, I really appreciated reading this paper and please find below my comments and suggestions:

1. Introduction: This section would need to be polished. Some words such as "salient" or "warranted" could be difficult for readers to understand. There are also some inappropriate uses of "accordingly". Moreover, it would improve the introduction if you talk in more details about circuits that are known to be involved in light, memory and stress. For example, SCN and hypothalamic related areas were shown to be important in light "alertness". You can also talk about the regions that we detected in the human studies you mentioned in line 118.
2. Statistics: please use non-parametric test (when applicable) as you cannot assume that your data have a Gaussian distribution as you have few data points.
3. Behaviour comment 1: I really appreciated that you have used two tests to assess memory but also stress. It really strengthens the results. However, I would feel more comfortable if you indicate that the two memory tests you have used are short-term spatial memory tests.
4. Behaviour comment 2: You performed your behaviour experiments during the light cycle (which would be similar to human sleep period). Could you explain this choice and what you would be the clinical implications?
5. In Fig. 1, adding the distance travelled in the Y maze and in the novel recognition task would give me more confidence that changes of light intensity do not affect locomotion.
6. For Fig 2a1 to h2: SCN, VLPO and DR regions were implicated in the past in light "alertness" (Lok, Smolders et al. 2018). Did you look at the c-Fos staining in those regions? If yes, are they changed?
7. In Fig 2 f1, the background is much higher compared to f2. Do you have an explanation for that? Did you use the same settings on the microscope?
8. In Fig 2, the panel a1 to h2 is not very clear. It is difficult to know which image belongs to which brain regions. Could you use space and maybe make it color-coded to make it clearer?
9. In Extended Data Fig. 2i and j, I noticed that GCaMP6m is not only expressed in DG but also in CA regions of hippocampus. Could you comment on this? What does make you think that you only recorded for the DG region?
10. In Extended Data Fig. 2, I have concern about the design of the calcium imaging experiment. Until now, you showed that you have enhanced memory retrieval when you have 50 lux of light during training and 400 lux during testing. However, during the calcium imaging, you change the light intensity during testing phase from 50 to 400 lux, instead of keeping it at 400 lux from the beginning. I think I understand the logistics and technical reasons why you chose this design but could you compare the signal with control animals that stayed in 50 lux? Can you comment on this?
11. In Fig 3g-n (helper virus experiment), this experiment does not eliminate the possibility that

CeA project directly to DG neurons. I would feel more confident if you could add control images where you don't inject the helper virus and you don't see fluorescence in DG.

12. In methods of c-Fos mapping, please indicate how long after the Y maze you collected the animals. This will help if other research groups would like to reproduce your data.

13. Discussion: adding a summary schematic would improve the discuss and help the readers to visualise your findings.

Reviewer #2 (Remarks to the Author):

The submitted manuscript by MengJuan Shanget et al investigated how the exposure to light can affect the behavior of mice in different models related to memory as well anxiety-related responses. The authors also investigated the neural circuits involved in the beneficial effect of the light (400LUX) which involves the central amygdala projections releasing CRF into Locus coeruleus and the release of norepinephrine into dentate gyrus.

The work is well designed, the questions and the way that the authors respond to them is very elegant and the using of cutting-edge techniques were well employed.

However, some points need to be addressed.

1) When the authors are talking about the anxiety-related behavior in mice they frequently refer to level of anxiety in the text. My question is what the authors want to state with it? Is it possible to measure levels of anxiety in an animal? Example, is the mouse more anxious in the end of the close arm of EPM than in the middle of the close arm, or near to the open arm?

2) About the cfos experiments the authors have analyzed any control area that is not regulate by alertness or cognition ?

3) Please, add references of previous work where the doses of antalarmin and propranolol were based.

4) The work has some limitations and would be good add a paragraph on the discussion part about the limitations of the work.

Reviewer #1 (Remarks to the Author):

In this manuscript, Professor Xing and colleagues have shown that short-term spatial memory in mice is improved when the animals are trained using a white light intensity of 50 lux but tested using 400 lux white light. However, those conditions induce some anxiety which is argued by the author to be the optimal amount of stress as higher light intensities during testing induce higher stress and do not enhance short-term memory retrieval in the Y maze compared to the control condition of 50 lux. In addition, the authors have tried to identify the brain circuits that are involved in this enhancement of short-term memory and they showed that CeA-LC-DG circuit is playing a crucial role using c-fos staining, calcium imaging and chemogenetics. They were also able to show that modulation of CRH+ neurons of CeA that are projecting to LC and modulation of TH+ neurons of LC that are projecting to DG impair the enhancement of the spatial memory when using 400 lux of white light during the test phase. Moreover, activation of CeA neurons projecting to LC increases the activity of CaMKIIa+ neurons in the DG. The authors, therefore, suggest a mechanism for the spatial memory retrieval at 400 lux where CRH+ neurons in the CeA activate TH+ neurons in the LC, which themselves active CaMKIIa+ neurons in the DG.

I would like to thank the authors for sharing their work with us. The manuscript shows a huge amount of effort and dedication. Intensity of white light was shown in the past to affect learning and memory but also the level of anxiety, as mentioned by the authors in the introduction. However, very little is known about the precise neuronal mechanism of this phenomenon. This work is one the first insights showing regions or circuits involved in this process. Overall, I really appreciated ready this paper and please find below my comments and suggestions:

Response to reviewer's comments: We much appreciate for your acknowledging of our research work, and we improved this manuscript based on your thoughtful comments and suggestions.

1. Introduction: This section would need to be polished. Some words such as “salient” or “warranted” could be difficult for readers to understand. There are also some inappropriate uses of “accordingly”. Moreover, it would improve the introduction if you talk in more details about circuits that are known to be involved in light, memory and stress. For example, SCN and hypothalamic related areas were shown to be important in light “alertness”. You can also talk about the regions that we detected in the human studies you mentioned in line 118 (line 386) .

Response to reviewer's comments: Thank you for the suggestions, as required, the related contents were added (P3 Second paragraph, Line 53-59), and the introduction section has been revised as indicated in highlight. In which, “salient” was changed to “significant” (line 40); “warrented” was changed to “necessary” (line 52), “accordingly” was changed to “Nevertheless” or “Therefore” (line 49 and line 51).

2. Statistics: please use non-parametric test (when applicable) as you cannot assume that your data have a Gaussian distribution as you have few data points.

Response to reviewer's comments: Thank you for the constructive suggestions. After consultation

with a statistician, we first tested the normality of the data, and then selected the corresponding statistical analysis method. The test showed that most of the data conform to normal distribution and can be analyzed by parametric tests. Other data were analyzed using non-parametric tests when applicable (Fig 1c and d, Fig 4h1, Fig 6f, Extended data Fig. 4h2), and also indicated in each figure legend in a highlighted text (line 116-118, line 240, line 329-330). Results from new statistic analysis did not change our conclusion.

3. Behaviour comment 1: I really appreciated that you have used two tests to assess memory but also stress. It really strengthens the results. However, I would feel more comfortable if you indicate that the two memory tests you have used are short-term spatial memory tests.

Response to reviewer's comments: Thank you for the suggestion. We have revised accordingly (Page 4 line 63 and Page 5 Line 89 &103)

4. Behaviour comment 2: You performed your behaviour experiments during the light cycle (which would be similar to human sleep period). Could you explain this choice and what you would be the clinical implications?

Response to reviewer's comments: Thank you for raising this important issue. We think that because of the harsh environments, rodents in nature feed at night and sleep during the day. However, in the laboratory setting, the mice were fed, watered and changed bedding during the day, and slept at night without human interference after the lights were turned off. In such a comfortable living environment, where daytime is no longer a threat to lab mice or their natural bedtime, the animals are more active during the day. From this point of view, the pace of life of lab mice may become similar to that of humans, and what light might do to laboratory mice during the day becomes somewhat representative of what it does to humans.

On the other side, compared to the brain regions shown in human MRI studies, we observed that the activated brain regions in experimental mice are very similar to those in humans [1, 2], suggesting that the light response in mice under the current experimental conditions is similar to that in humans. In fact, human studies have reported that light at work improves cognition performance, and we've seen this in mice as well. To a certain extent, these mechanisms provide theoretical guidance for the measurement of light parameters to improve cognition in clinic. Therefore, we believe that the mechanism elaborated in this study can provide some evidence for human understanding of "optimal learning effect induced by moderate anxiety", and may provide some theoretical guidance for the formulation of clinical light parameters.

[1] Vandewalle G, Balteau E, Phillips C, et al. Daytime light exposure dynamically enhances brain responses. *Curr Biol*, 2006, 16: 1616-1621

[2] Vandewalle G, Schmidt C, Albouy G, et al. Brain responses to violet, blue, and green monochromatic light exposures in humans: Prominent role of blue light and the brainstem. *PLoS One*, 2007, 2: e1247

5. In Fig. 1, adding the distance travelled in the Y maze and in the novel recognition task would give me more confidence that changes of light intensity do not affect locomotion.

Response to reviewer's comments: In Figure 1c and Extended Figure 1c of new version, the

analysis of “distance travelled in the Y maze and in the novel recognition task” had been added. The change of light intensity did not affect locomotion. The corresponding content was added to the results section (P5 line 85-87).

Figure 1c was added for distance analysis.

6. For Fig 2a1 to h2: SCN, VLPO and DR regions were implicated in the past in light “alertness” (Lok, Smolders et al. 2018). Did you look at the c-Fos staining in those regions? If yes, are they changed?

Response to reviewer’s comments: Thank you for the question. We didn’t look at the c-Fos staining in SCN, VLPO and DR regions before the first version of manuscript was submitted. However, as you mentioned, these areas are implicated in light “alertness”, so we completed relevant c-fos staining experiments. There was no difference in c-fos expression in these three regions under illumination conditions of 50 lux and 400lux (as shown in the figures below), so they are not considered in the current study.

7. In Fig 2 f1, the background is much higher compared to f2. Do you have an explanation for that? Did you use the same settings on the microscope?

Response to reviewer's comments: Thank you for the critical question. When c-fos staining was performed, we tried to control similar experimental conditions and microscope sampling conditions. In Fig.2f, we focused on spot c-fos staining, so the level of background staining might be ignored when similar areas were guaranteed. In the process of this revision, we carried out another batch of c-fos staining and image collection operations, and replaced the original image in Fig 2f in this version of the manuscript.

8. In Fig 2, the panel a1 to h2 is not very clear. It is difficult to know which image belongs to which brain regions. Could you use space and maybe make it color-coded to make it clearer?

Response to reviewer's comments: Thanks for the comments. As suggested, we have used space and made it color-coded for the panels a1 to h2 in figure 2 of the newly uploaded version of manuscript.

In the flowing Figure 2, Fig.2f was changed in response to Question 7; Usage of space and color-coded were in response to Question 8

i

9. In Extended Data Fig. 2i and j, I noticed that GCaMP6m is not only expressed in DG but also in CA regions of hippocampus. Could you comment on this? What does make you think that you only recorded for the DG region?

Response to reviewer's comments: Thanks for the critical question. Due to the projection between different hippocampal subregions, a small amount of AAV-hSyn-GCaMP6m virus injected into DG may be absorbed by the axonal terminals of neurons projected from other region to DG. Therefore, a small number of neurons in CA1 region may also be infected by the virus and express green fluorescence. In this experiment, since the optical fiber was placed about 150 μm above DG and basically below CA1 during calcium signal recording, it was unlikely that the virus

fluorescence in CA1 region could be recorded by optical fiber. In addition, theoretically, only the excited light located about 150-200 μ m below the fiber can be effectively received, so the calcium signal in this experiment can be well guaranteed to come from the DG region. However, the original diagram is very ambiguous, so we have replaced it with another diagram (see in the following Extended Data Fig. 2i and j) that is also more limited in virus expression.

10. In Extended Data Fig. 2, I have concern about the design of the calcium imaging experiment. Until now, you showed that you have enhanced memory retrieval when you have 50 lux of light during training and 400 lux during testing. However, during the calcium imaging, you change the light intensity during testing phase from 50 to 400 lux, instead of keeping it at 400 lux from the beginning. I think I understand the logistics and technical reasons why you chose this design but could you compare the signal with control animals that stayed in 50 lux? Can you comment on this?

Response to reviewer's comments: Thanks for the good question. The purpose of this experiment

section was to qualitatively analyze the results of c-fos mapping. We then examined whether the neurons in the corresponding brain regions show increase of fluorescence due to activation by different intensities of light, which provide further evidences supporting the involvement of these three brain regions in changes of animal behaviors. Indeed, the fluorescence intensity of the three brain regions changed when light intensity increased from 50lux (background) to 400lux.

11. In Fig 3g-n (helper virus experiment), this experiment does not eliminate the possibility that CeA project directly to DG neurons. I would feel more confident if you could add control images where you don't inject the helper virus and you don't see fluorescence in DG.

Response to reviewer's comments: We are a little bit unclear about this question “.....don't see fluorescence in DG” in which the fluorescence is in DG but not in CeA, since reviewer want to eliminate the possibility that CeA project directly to DG neurons. As suggested, we injected RV virus (RV-ENVA-ΔG-dsRed) into the DG region without injecting the helper virus (AAV2/9-hSyn-EGFP-2a-TVA-2a-RVG) in the LC region. Tissue was sectioned to detect the expression of virus in the DG, CeA and LC regions. The rabies virus retrograde transsynaptic tracer system are known to be fully expressed and infected neurons only in the presence of helper viruses. As shown in the following figures, RV did not infect the DG neurons without co-injection of helper virus, and only the fluorescence signal of the viral vector itself was detected in the DG region, similar to the effect of fluorescent dyes as shown in the following figure A. Similarly, there was no red fluorescence detected in LC (figure B), suggesting the rabies virus alone did not infect its upstream CeA neurons, which is consistent with figure C. In this experiment, no direct projections from the CeA to DG neurons was observed. Actually, using the retrograde tracing strategy as shown in figure 3 a-f, we didn't find retrograde labeling in the CeA after viruses was injected into DG, suggesting that there 's no direct projection from CeA to DG.

12. In methods of c-Fos mapping, please indicate how long after the Y maze you collected the animals. This will help if other research groups would like to reproduce your data.

Response to reviewer's comments: Thanks for the suggestions. The time of collecting tissue for C-fos mapping was 1.5 hours after Y-maze test. Relevant information has been added in the method section of file "Online Methods" (Page 3 line 65 to 66)

13. Discussion: adding a summary schematic would improve the discuss and help the readers to visualise your findings.

Response to reviewer's comments: Thanks for the suggestion. The following summary schematic is added in discussion section as Figure 8.

Reviewer #2 (Remarks to the Author):

The submitted manuscript by MengJuan Shanget et al investigated how the exposure to light can affect the behavior of mice in different models related to memory as well anxiety-related responses. The authors also investigated the neural circuits involved in the beneficial effect of the light (400LUX) which involves the central amygdala projections releasing CRF into Locus coeruleus and the release of norepinephrine into dentate gyrus. The work is well designed, the questions and the way that the authors respond to them is very elegant and the using of cutting-edge techniques were well employed. However, some points need to be addressed.

Response to reviewer's comments: We are very grateful for your recognition of our work and we improved the manuscript based on your thoughtful comments and suggestions.

1). When the authors are talking about the anxiety-related behavior in mice they frequently refer to level of anxiety in the text. My question is what the authors want to state with it? Is it possible to measure levels of anxiety in an animal? Example, is the mouse more anxious in the end of the close arm of EPM than in the middle of the close arm, or near to the open arm?

Response to reviewer's comments: Thank you for the good question. Mice spent less time in the center area of the open field and the open arm of the elevated cross maze, indicating an elevated anxious status. Therefore, this residence time represents an objective parameter for evaluating animal's anxious behavior. We agree that "anxiety level" is not a professional term, and we used it

to convey our idea that moderate anxiety promotes memory retrieval. The anxiety level is actually a relative description, which may not seem normative. We look forward to finding better term in the future or from your kind suggestion.

2). About the c-fos experiments the authors have analyzed any control area that is not regulate by alertness or cognition ?

Response to reviewer's comments: In Fig 2, we analyzed the c-fos expression in eight brain regions, five of which were associated with alertness and cognition, while the other three were not. These 3 areas include posterolateral cortical amygdala (PLCO), dorsomedial hypothalamic nucleus (DMH) and PrC. In which, the PLCO is known to be involved in innate, odor-driven behavioral response[3, 4]. The DMH is involved in feeding, drinking, and body weight regulation[5]. The PrC may well be involved directly in organizing somatomotor, autonomic, and analgesic responses associated with defensive behavior [6]. These three regions had a high basal expression of c-fos, but no difference in c-fos expression between 400 lux and 50 lux groups.

[3] Ubeda-Banon I, Novejarque A, Mohedano-Moriano A, et al. Projections from the posterolateral olfactory amygdala to the ventral striatum: Neural basis for reinforcing properties of chemical stimuli. *BMC Neurosci*, 2007, 8: 103

[4] Root CM, Denny CA, Hen R, et al. The participation of cortical amygdala in innate, odour-driven behaviour. *Nature*, 2014, 515: 269-273

[5] Bellinger LL, Bernardis LL. The dorsomedial hypothalamic nucleus and its role in ingestive behavior and body weight regulation: Lessons learned from lesioning studies. *Physiol Behav*, 2002, 76: 431-442

[6] Canteras NS, Goto M. Connections of the precommissural nucleus. *J Comp Neurol*, 1999, 408: 23-45

3). Please, add references of previous work where the doses of antalarmin and propranolol were based.

Response to reviewer's comments: The added references can be found in the results section and reference list (P18 Line 307, No28: PMID 26212712; P20 Line 337, No 29: PMID 27930324)

4). The work has some limitations and would be good add a paragraph on the discussion part about the limitations of the work.

Response to reviewer's comments: Thanks for the suggestion. We have added some discussions to acknowledge these limitations (last paragraph of this section, P26 line 455-464).

REVIEWERS' COMMENTS:

Reviewer #1 (Remarks to the Author):

Thank you to the authors for taking into consideration the reviewers' comments. Overall, I am satisfied with the revised manuscript. I only have a couple of minor comments:

1. Introduction: Thank you very for adding the paragraph from Line 53 to 58. It really helped with the understanding of the concepts. However, please be careful with the flow of the introduction. It does not always make sense. For example, the Lines 59 to 65 are difficult to follow. The sentence in Line 66-68 is maybe too long and could be improved. Finally, references are missing for Line 53-54 (about anxiety, evolution and alertness/avoid danger) and for Line 61 (about aversive stimulus inducing anxiety).
2. Missing scale bar in Fig. 4g1, Extended Data Fig. 4g1 and Extended Data Fig 6i.
3. Please change "corticotrophin" in Line 296 to "corticotropin". CRF stands for Corticotropin-releasing Factor.
4. Please add more details on the viruses used for the neurotransmitter sensors with references.

Reviewer #2 (Remarks to the Author):

The authors responded to my all questions and made great improvement on the manuscript. This is a really good work and very well designed. I would like to congratulate Professor Xing and collaborators for this work.

About the terminology of anxiety level what about if instead of talk about the anxiety we talk about of the threat. Example, instead of saying that the level of anxiety when the mice was on the center of the open field is high we can say that the animal is more near/exposed to the threat. It may be a good way to not infer about how the animal would actually be feeling. I am also not saying that it needs be done in this manuscript but maybe that's a good start point to the group discuss about these things for the next works.
Congratulations to the team again!

Reviewer #1 (Remarks to the Author):

Thank you to the authors for taking into consideration the reviewers' comments. Overall, I am satisfied with the revised manuscript. I only have a couple of minor comments:

1. Introduction: Thank you very for adding the paragraph from Line 53 to 58. It really helped with the understanding of the concepts. However, please be careful with the flow of the introduction. It does not always make sense. For example, the Lines 59 to 65 are difficult to follow. The sentence in Line 66-68 is maybe too long and could be improved. Finally, references are missing for Line 53-54 (about anxiety, evolution and alertness/avoid danger) and for Line 61 (about aversive stimulus inducing anxiety).

Response to reviewer's comments: Thanks for the questions. We have rephrased the narrative to make it easier to follow (line 60-66). The long sentence in line 66-68 has also been simplified. The missing references have been added at the appropriate location (line 56, line 63)

2. Missing scale bar in Fig. 4g1, Extended Data Fig. 4g1 and Extended Data Fig 6i.

Response to reviewer's comments: The missing scale bar has been added in the new version of the figures.

3. Please change "corticotrophin" in Line 296 to "corticotropin". CRF stands for Corticotropin-releasing Factor.

Response to reviewer's comments: Thanks for the correction, we have revised it as suggested (line 209).

4. Please add more details on the viruses used for the neurotransmitter sensors with references.

Response to reviewer's comments: The reference has been added (line 215 and line 232).

Reviewer #2 (Remarks to the Author):

The authors responded to my all questions and made great improvement on the manuscript. This is a really good work and very well designed. I would like to congratulate Professor Xing and collaborators for this work.

About the terminology of anxiety level what about if instead of talk about the anxiety we talk about of the threat. Example, instead of saying that the level of anxiety when the mice was on the center of the open field is high we can say that the animal is more near/exposed to the threat. It may be a good way to not infer about how the animal would actually be feeling. I am also not saying that it needs be done in this manuscript but maybe that's a good start point to the group discuss about these things for the next works. Congratulations to the team again!

Response to reviewer's comments: We much appreciate for your acknowledging of our research work and the kind suggestions.